PREPARED FOR SUBMISSION TO JHEP

# Holography for bulk states in 3D quantum gravity

**Joris Raeymaekers and Gideon Vos**

*CEICO, Institute of Physics of the ASCR, Na Slovance 2, 182 21 Prague 8, Czech Republic.*

*E-mail:* joris@fzu.cz, vos@fzu.cz

ABSTRACT: In this work we discuss the holographic description of states in the Hilbert space of (2+1)-dimensional quantum gravity, living on a time slice in the bulk. We focus on pure gravity coupled to pointlike sources for heavy spinning particles. We develop a formulation where the equations for the backreacted metric reduce to two decoupled Liouville equations with delta-function sources under pseudosphere boundary conditions. We show that both the semiclassical wavefunction and the gravity solution are determined by a universal object, namely a classical Virasoro vacuum block on the sphere. In doing so we derive a version of Polyakov's conjecture, as well as an existence criterion, for classical Liouville theory on the pseudosphere. We also discuss how some of these results are modified when considering closed universes with compact spatial slices.

## 1 Introduction and summary

In its most common formulation, the AdS/CFT correspondence [1] allows for the computation of correlation functions in a strongly coupled CFT from AdS gravity in the bulk with sources placed on the boundary [2, 3]. Though the CFT description is believed to capture the full gravitational Hilbert space, it is far from obvious how quantum gravity states, defined on some time slice $\Sigma$, are described in CFT terms. It would therefore be desirable to have a more direct holographic dictionary relating quantum states in the bulk to CFT quantities. Such a reformulation of holography was referred to as 'CFT/AdS' by H. Verlinde in an inspiring lecture [4]. It would allow one to address important issues such as the description of local excitations or of the black hole interior. These are closely related to the question of holographic emergence of bulk locality which is a subject of intensive research.

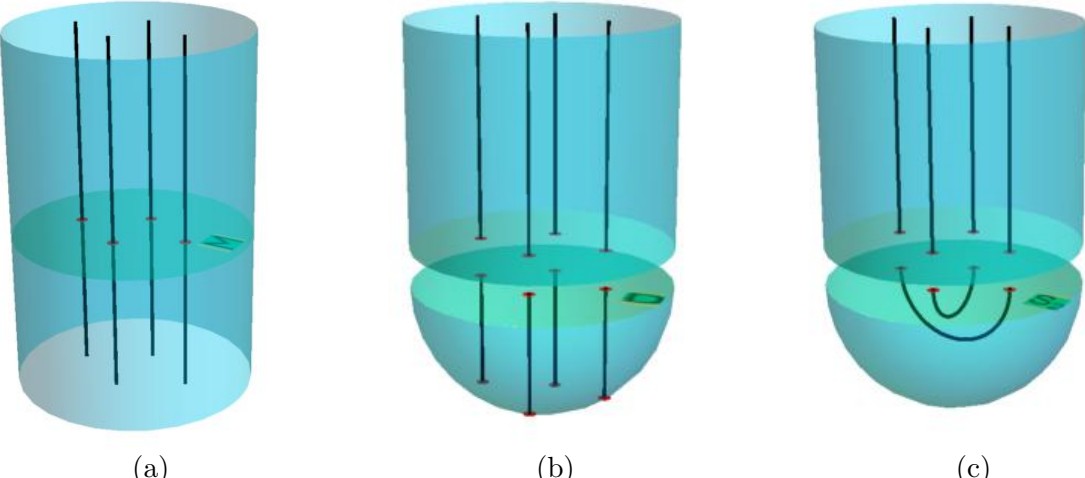

<table>
<tr><td>(a)</td><td>(b)</td><td>(c)</td></tr>
</table>

**Figure 1**. (a) A collection of point particles in AdS piercing an initial time slice $\Sigma$. (b) When $\Sigma$ is a disk, we prepare the state with a path integral over 3-geometries with particles emerging from the boundary in the past. (c) When $\Sigma$ is a two-sphere, we compute a path integral over 3-geometries without past boundary.

In this work we will address the holographic description of certain bulk states in 2+1 dimensional quantum gravity. Our scope will be rather modest, focusing on states containing a number of point-like particles but no black holes. Though we will make heavy use of the quasi-topological nature of (2+1)D gravity and of its Chern-Simons description, one might hope that some of our results, such as the connection to CFT conformal blocks that we will uncover, generalize to higher dimensions as well. A different approach to bulk state holography was proposed in [5].

More precisely, we will consider states in the bulk which contain a number of massive, spinning particles in AdS in a point-like limit. Their worldlines pierce an initial time slice $\Sigma_0$ with the topology of a disk, as sketched schematically in Figure 1(a).

The particles are taken to be sufficiently heavy to backreact on the geometry and produce a multi-centered AdS solution.

At the classical level, the state of the gravitational field we are interested in consists of initial data on $\Sigma_0$ which satisfy the Hamiltonian and momentum constraints, and has the appropriate behavior near the particle sources. We will introduce a parametrization of (2+1)D gravity where these initial value constraints reduce to two decoupled Liouville equations with delta-function sources

$$\partial_z \partial_{\bar{z}} \Phi + e^{-2\Phi} = \pi \sum_i \alpha_i \delta^{(2)}(z - z_i), \qquad \partial_z \partial_{\bar{z}} \tilde{\Phi} + e^{-2\tilde{\Phi}} = \pi \sum_i \tilde{\alpha}_i \delta^{(2)}(z - z_i) \qquad (1.1)$$

Here, complex coordinates range over the upper half plane, with $z_i$ the locations where the particle worldlines pierce the initial slice $\Sigma_0$, and $\alpha_i = 2G(m_i + s_i)$, $\tilde{\alpha}_i = 2G(m_i - s_i)$, with $m_i$ and $s_i$ the particle masses and helicities respectively. The Liouville fields $\Phi$ and $\tilde{\Phi}$ obey the pseudosphere boundary conditions of [6] on the real line. The metric on the initial value surface is then given in terms of these fields as

$$ds^2_{\Sigma_0} = \left(e^{-\Phi} + e^{-\tilde{\Phi}}\right)^2 dz d\bar{z} - \left(\text{Im}\left(\partial_z(\Phi - \tilde{\Phi})dz\right)\right)^2. \qquad (1.2)$$

To find the full (2+1)D metric one needs to integrate a system of first order time evolution equations spelled out in (2.36) below. While generically quite complicated, the time dependence simplifies in cases where all the particles are either 'chiral' or 'antichiral', meaning

that $\alpha_i \tilde{\alpha}_i = 0$, $\forall i$. The full 2+1 dimensional metric is in that case

$$ds^2 = \left| e^{-\Phi(z_+,\bar{z}_+)}dz_+ + e^{-\tilde{\Phi}(z_-,\bar{z}_-)}dz_- \right|^2 - \left( \text{Im} \left( \partial_{z_+}\Phi(z_+,\bar{z}_+)dz_+ - \partial_{z_-}\tilde{\Phi}(z_-,\bar{z}_-)dz_- \right) \right)^2,$$

where $z_\pm = z \pm t$. This class of metrics generalizes that studied in [7], where purely chiral solutions with $\tilde{\alpha}_i = 0, \forall i$ were considered.

On the quantum level, a useful way to represent a bulk state is as a Hartle-Hawking path integral over three-surfaces with the time slice $\Sigma_0$ as boundary. Our goal is to give a direct CFT interpretation of this object. To prepare the multi-particle state on $\Sigma_0$ we perform a path integral on a 3-manifold $X$ ending on $\Sigma_0$, containing particle worldlines emerging from the boundary in the past and terminating on $\Sigma_0$ (see Figure 1(b)). In the leading approximation, we have to evaluate the classical on-shell action on $X$ with appropriate boundary terms. In doing so we will find a close connection between the wavefunction and a holomorphic Virasoro conformal block on the sphere. We will find a relation of the form

$$\Psi_{HH}[(z_i,\bar{z}_i),\alpha_i,\tilde{\alpha}_i] \sim N(\alpha_i,\tilde{\alpha}_i) \exp\left( -\frac{k}{4}(F_0(z_i,\bar{z}_i;\alpha_i) + F_0(z_i,\bar{z}_i;\tilde{\alpha}_i) \right) \qquad (1.3)$$

Here $N(\alpha_i,\tilde{\alpha}_i)$ is a normalization factor independent of the $z_i$, and $F_0(z_i,\bar{z}_i;\alpha_i)$ is a classical Virasoro block for a $2n$-point correlator on the sphere, with identical operators inserted in each $z_i$ and its image point $\bar{z}_i$. The subscript in $F_0$ indicates that it is a vacuum block in the channel where the operators in image points are fused pairwise, as illustrated in Figure 3. The fact that, in the semiclassical approximation, the features of the state are captured by a universal object insensitive to the details of the UV completion, is a concrete realization of general expectations [8]. In the process of deriving the relation (1.3), we will also derive a version of Polyakov's conjecture [9, 10] for Liouville theory on the pseudosphere, and give an existence criterion for Liouville solutions on the pseudosphere in terms of a certain reflection property of the classical block $F_0$.

The classical block $F_0$ does not only determine the Hartle-Hawking wavefunction, but also the full 2+1 dimensional metric. The Liouville solution in (1.1) is of the form

$$e^\Phi = i(\psi_1 \bar{\psi}_2 - \bar{\psi}_1 \psi_2) \qquad (1.4)$$

where $\psi_1(z)$ and $\psi_2(z)$ are the two independent solutions (with unit Wronskian, i.e. $\psi_1'\psi_2 - \psi_1\psi_2' = 1$) to an ordinary differential equation whose coefficients are determined by $F_0$, namely

$$(\partial_z^2 + T(z))\psi = 0, \qquad (1.5)$$

with

$$T(z) = \sum_{i=1}^n \left( \frac{\alpha_i(2-\alpha_i)}{4(z-z_i)^2} + \frac{\alpha_i(2-\alpha_i)}{4(z-\bar{z}_i)^2} + \frac{\partial_{z_i}F_0}{z-z_i} + \frac{\overline{\partial_{z_i}F_0}}{z-\bar{z}_i} \right). \qquad (1.6)$$

Our approach extends in principle to initial slices with a different topology. It is interesting to consider the case where $\Sigma_0$ is compact without boundary. Here we expect to find a CFT description of closed universes with negative cosmological constant which

contain some point particles. We will focus on the case where $\Sigma_0$ has spherical topology, and consider a Hartle-Hawking path integral over a ball containing worldlines of particles pair-created in the past (see Figure 1(c)). In contrast to the case with conformal boundary, the wavefunction and backreacted gravity solution now depend on dynamical CFT data, rather than on conformal kinematics alone. Indeed, we will find relations similar to (1.6) and (1.3) above, where the vacuum block $F_0$ is replaced by a specific non-vacuum block

$$F(z_i, \alpha_i, \beta_I^*), \tag{1.7}$$

where $\beta_I^*$ label the exchanged conformal families. These depend implicitly on the insertion points, $\beta_I^* = \beta_I^*(z_i, \bar{z}_i)$, in a highly complicated manner. In fact, to determine them one needs dynamical information in the form of in the large $c$ behavior of the Liouville three-point functions.

This paper is organized as follows. In Section 2 we introduce a convenient parametrization of (2+1)D gravity in terms of two auxiliary 2D hyperbolic metrics. A judicious choice of gauge allows us to rephrase the backreaction of spinning point particles as a pair of decoupled inhomogeneous Liouville equations. In Section 3 we apply our formalism to asymptotically AdS spacetimes and derive a connection between the Hartle-Hawking-like wavefunction, the on-shell Liouville action on a pseudosphere, and a classical Virasoro block on the sphere. For this purpose we derive a version of Polyakov's conjecture and give a new existence criterion for Liouville theory on the pseudosphere. We also discuss some special cases and explicit examples. In Section 4 we discuss the modifications occurring when considering closed universes with compact spatial slices. We end with some open problems and future directions.

## 2 'Doubly hyperbolic' parametrization of 2+1 dimensional gravity

In this section we introduce a convenient parametrization of (2+1)D gravity. It relies heavily on the formulation of the theory as a Chern-Simons theory with gauge group $SL(2, \mathbb{R}) \times SL(2, \mathbb{R})$. The constraint equation of $SL(2, \mathbb{R})$ Chern-Simons theory, which states that the field strength should vanish on spacelike slices $\Sigma$, can be interpreted as the 2D Euclidean Einstein equation with negative cosmological constant [11]. In this way we obtain a reformulation of (2+1)D gravity as a time evolution of two auxiliary hyperbolic metrics. Perhaps unsurprisingly, the problem of of backreacting point-particle sources then reduces to solving the Liouville equation on $\Sigma$ with delta-function sources.

### 2.1 Chern-Simons description

Our starting point is the Chern-Simons formulation of (2+1)D gravity:

$$S = S_{CS}[A] - S_{CS}[\tilde{A}], \qquad S_{CS}[A] = \frac{k}{4\pi} \text{tr} \int_{\mathcal{M}} \left( A \wedge dA + \frac{2}{3} A \wedge A \wedge A \right), \tag{2.1}$$

where $A$ and $\tilde{A}$ are potentials for the gauge group $SL(2,\mathbb{R})$. The gauge potentials are related to the vielbein $E$ and spin connection $\Omega$ as[1]

$$A = \Omega + E, \qquad \tilde{A} = \Omega - E. \tag{2.2}$$

The generators of $sl(2,\mathbb{R})$ satisfy the commutation relations $[L_m, L_n] = (m-n)L_{m+n}$, $m, n = 0, \pm 1$. The trace in (2.1) will, for definiteness, be taken in the 2-dimensional representation[2], where $\mathrm{tr} L_a L_b = \frac{1}{2}\eta_{ab}$. The Chern-Simons level $k$ is then related to the Brown-Henneaux central charge [12] as

$$c := \frac{3}{2G} = 6k. \tag{2.4}$$

In the present Section take the manifold $\mathcal{M}$ on which the Chern-Simons theory is defined to be of the form $\mathbb{R} \times \Sigma$, where the real line is the time direction and $\Sigma$ is a Riemann surface. For concreteness we will use $t \in \mathbb{R}$ as the time coordinate and choose a local complex coordinate $z$ on $\Sigma$. To clarify the canonical structure, it is useful to denote the spatial projections of $A, \tilde{A}$ by a hat,

$$\hat{A} \equiv A_z dz + A_{\bar{z}} d\bar{z}, \qquad \hat{\tilde{A}} \equiv \tilde{A}_z dz + \tilde{A}_{\bar{z}} d\bar{z} \tag{2.5}$$

and similarly introduce a spatial exterior derivative

$$\hat{d} = dz \partial_z + d\bar{z} \partial_{\bar{z}}. \tag{2.6}$$

The canonical structure of the Chern-Simons action is clarified by rewriting it, up to a total derivative, as

$$S[A] = -\frac{k}{4\pi} \mathrm{tr} \int_{\mathbb{R}\times\Sigma} dt \wedge \left( \hat{A} \wedge \dot{\hat{A}} - 2 A_t \hat{F} \right). \tag{2.7}$$

The time component $A_t$ is not a dynamical variable but a Lagrange multiplier enforcing the *constraint equation*

$$\hat{F} = \hat{d}\hat{A} + \hat{A} \wedge \hat{A} = 0, \tag{2.8}$$

The remaining (complex) equation of motion

$$F_{tz} = 0 \tag{2.9}$$

is a *dynamical equation* describing the time evolution of the dynamical variables $\hat{A}$.

---

[1]Throughout, we work in units where the AdS radius is set to one.

[2]For concreteness we can take

$$L_0 = \frac{1}{2}\begin{pmatrix} 1 & 0 \\ 0 & -1 \end{pmatrix}, \qquad L_1 = \begin{pmatrix} 0 & -1 \\ 0 & 0 \end{pmatrix}, \qquad L_{-1} = \begin{pmatrix} 0 & 0 \\ 1 & 0 \end{pmatrix}. \tag{2.3}$$

## 2.2 Choice of gauge

In what follows, we will use the gauge freedom to work in a generalization of the temporal gauge $A_t = \tilde{A}_t = 0$. It involves choosing two smooth vector fields $V$ and $\tilde{V}$ which are tangent to $\Sigma$ at all times, i.e.

$$V = V^z \partial_z + V^{\bar{z}} \partial_{\bar{z}}, \qquad \tilde{V} = \tilde{V}^z \partial_z + \tilde{V}^{\bar{z}} \partial_{\bar{z}}. \tag{2.10}$$

We then impose the gauge conditions

$$A_t = -i_V \hat{A}, \qquad \tilde{A}_t = -i_{\tilde{V}} \hat{\tilde{A}}. \tag{2.11}$$

In the nomenclature of [13], this is a Lagrange multiplier gauge which, for $V = \tilde{V} = 0$, reduces to the temporal gauge. We will keep the vector fields $V$ and $\tilde{V}$ general for the moment; in what follows they will be chosen judiciously in order to simplify the problem. We should note that, if $V$ and $\tilde{V}$ coincide on some locus, then the vielbein $E$ (see (2.2)) degenerates there and we have a coordinate singularity. Below, when adding particle sources, this will actually happen on the isolated worldlines, reflecting the conical singularity of the metric there [14, 15].

Using (2.11) to eliminate $A_t$, the equations for the dynamical variables reduce to

$$\hat{F} = 0, \qquad\qquad\qquad \hat{\tilde{F}} = 0 \tag{2.12}$$

$$\partial_t \hat{A} = -\hat{\mathcal{L}}_V \hat{A}, \qquad\qquad\qquad \partial_t \hat{\tilde{A}} = -\hat{\mathcal{L}}_{\tilde{V}} \hat{\tilde{A}} \tag{2.13}$$

The first line contains the constraint equations on the time slice $\Sigma$, while on the second line we have the dynamical equations determining the time evolution.

The gauge choice (2.11) has residual symmetries stemming from parameters satisfying[3]

$$\dot{\Lambda} = -\hat{\mathcal{L}}_V \Lambda, \qquad \dot{\tilde{\Lambda}} = -\hat{\mathcal{L}}_{\tilde{V}} \tilde{\Lambda}. \tag{2.14}$$

The parameter $\Lambda$ can be chosen arbitrarily on specific time slice, say at $t = 0$, which can be used to bring $\hat{A}, \hat{\tilde{A}}$ to a convenient form there.

## 2.3 'Doubly hyperbolic' variables

We now introduce a convenient parametrization for the spatial connections $\hat{A}$ and $\hat{\tilde{A}}$. This parametrization naturally arises from the relation [11] between flat $SL(2, \mathbb{R})$ connections and constant negative curvature metrics on $\Sigma$.

Concretely, we parametrize $\hat{A}, \hat{\tilde{A}}$ as

$$\hat{A} = e L_1 - \bar{e} L_{-1} + i\omega L_0, \qquad \hat{\tilde{A}} = -\tilde{e} L_1 + \bar{\tilde{e}} L_{-1} + i\tilde{\omega} L_0, \tag{2.15}$$

---

[3]Here and in what follows, we define the spatial Lie derivative $\hat{\mathcal{L}}_V = i_V \hat{d} + \hat{d} i_V \mathrm{m}$ to act on form indices only, and not on Lie algebra indices.

where the one-forms $e$ and $\bar{e}$ are related by complex conjugation and $\omega$ is real[4], and similarly for the quantities with a tilde. The constraint equations (2.12) reduce to

$$\hat{d}e - i\omega \wedge e = 0, \tag{2.16}$$

$$\hat{d}\omega + 2ie \wedge \bar{e} = 0, \tag{2.17}$$

and similarly for $\tilde{e}, \tilde{\omega}$. These equations state that, at each time $t$, the two auxiliary metrics on $\Sigma$

$$ds_2^2 = e\bar{e}, \qquad d\tilde{s}_2^2 = \tilde{e}\bar{\tilde{e}} \tag{2.18}$$

have constant negative curvature, while $\omega, \tilde{\omega}$ the role of the associated spin connection one-forms. The latter are determined algebraically in terms of $e, \tilde{e}$ and their spatial derivatives through (2.16).

The dynamical equations (2.13) become

$$\dot{e} = -\hat{\mathcal{L}}_V e, \qquad \dot{\tilde{e}} = -\hat{\mathcal{L}}_{\tilde{V}}\tilde{e}. \tag{2.19}$$

The dynamical equation for the spin connection, $\dot{\omega} = -\hat{\mathcal{L}}_V \omega$ is automatically satisfied when expressing $\omega$ in terms of $e$ and using (2.19).

The actual $(2+1)$D gravity metric $ds^2 = \mathrm{tr}(A - \tilde{A})^2/2$ takes the form

$$ds^2 = \left| (i_V e + i_{\tilde{V}}\tilde{e})\, dt - e - \tilde{e} \right|^2 - \frac{1}{4}\left( (i_V \omega - i_{\tilde{V}}\tilde{\omega})\, dt - \omega + \tilde{\omega} \right)^2. \tag{2.20}$$

It is generically related to the two auxiliary constant curvature metrics (2.18) in a nontrivial way. In the special left-right symmetric case where $e = \tilde{e}$, the metric on $\Sigma$ is simply proportional to the auxiliary constant negative curvature metrics (2.18), $ds_\Sigma^2 = 4ds_2^2 = 4d\tilde{s}_2^2$. Furthermore, if $V = -\tilde{V}$, the $(2+1)$D metric in this class is static. In Section (3.6) we will discuss another class of solutions, obtained as a certain scaling limit, which are fibrations over a 2D base with hyperbolic metric $ds_2^2$.

**More on the phase space**

While the parametrization of the gauge potentials (2.15,2.28,2.29) may seem cumbersome from the point of view of $(2+1)$D metric variables, it gives a useful 'left-right factorized' description of the gravity phase space which, following [16], facilitates the connection to 2D CFT. Indeed, from the above parametrization we see that the space of flat $SL(2, \mathbb{R})$ gauge potentials (2.15) for which the 2D vielbein $(e, \bar{e})$ is invertible[5], modulo gauge transformations, is given by the space of constant negative curvature vielbeins modulo diffeomorphisms and local Lorentz transformations. In other words, this phase space is, schematically,

$$\frac{\text{metrics on } \Sigma}{\mathrm{Diff}_0(\Sigma)\ \times \mathrm{Weyl}}. \tag{2.21}$$

---

[4]Note that with these reality conditions $\hat{A}$ and $\hat{\tilde{A}}$ actually take values in $su(1,1)$. A similarity transformation with $e^{i\pi(L_1 - L_{-1})/4}$ would yield $sl(2,\mathbb{R})$-valued potentials, though here we will stick with the simpler form (2.15).

[5]As shown in [17],[18], the full moduli space of flat $SL(2, \mathbb{R})$ connections consists of several components, one of which is the Teichmüller space and which is picked out by the restriction to inverible vielbeins.

If $\Sigma$ is of genus $g$ and has $b$ boundaries, this space can be identified as the Teichmüller space $\mathcal{T}(g,b)$. From the 2+1 decomposition of the the Chern-Simons action (2.7) we see that the symplectic form on the phase space is given by

$$\omega_{CS} = -\frac{k}{4\pi}\text{tr}\int_\Sigma \delta\hat{A}\wedge\delta\hat{A}, \qquad (2.22)$$

which can be shown to be equivalent to the standard Weil-Petersson symplectic form on $\mathcal{T}(g,b)$ [11]. Combining with the analogous $\tilde{\hat{A}}$ phase space we conclude that the phase space of (2+1)D gravity is, locally[6],

$$\mathcal{T}(g,b)\times\mathcal{T}(g,b). \qquad (2.23)$$

A different parametrization of the field space, which is more natural in metric variables, describes the phase space as the cotangent bundle of a single copy of the Teichmüller space, $T^*(\mathcal{T}(g,b))$. These two descriptions are equivalent in a nontrivial way, as shown in [19, 20].

We should remark that, if the boundaries are asymptotic boundaries, which will be the case of interest for us, the diffeomorphisms $\text{Diff}_0(\Sigma)$ in (2.21) are defined to act trivially at infinity, and the resulting Teichmüller space is in fact infinite dimensional [16]. It captures 'boundary graviton' excitations which we will describe more explicitly in Section 3. For example, for $g = 0, b = 1$, it can be shown that $\mathcal{T}(0,1)$ can be identified with $\text{Diff}(S^1)/SL(2,\mathbb{R})$ [16, 21]. The Weyl-Petersson symplectic structure reduces to the Kirillov-Kostant [22] structure on the vacuum Virasoro coadjoint orbit, and it's quantization leads to the Virasoro vacuum representation [23].

**Residual gauge fixing: Fefferman-Graham gauge**

The residual gauge freedom (2.14) can be used to bring the spatial connections $\hat{A}, \tilde{\hat{A}}$ in a desired form on the $t = 0$ initial slice which we will denote as $\Sigma_0$. Before discussing the conformal gauge, which will be used in the rest of the paper, it is instructive to first see how the standard 'Fefferman-Graham gauge' for asymptotically AdS spacetimes fits in our parametrization. This gauge leads to the (2+1)D metric in the Banados form of [24], which is an all-order version of the Fefferman-Graham expansion.

The Fefferman-Graham gauge corresponds to taking the 2D vielbeins to be of the form

$$e = \frac{dz}{2y} - \frac{y}{2}T(x)dx, \qquad \tilde{e} = \frac{dz}{2y} - \frac{y}{2}\tilde{T}(x)dx. \qquad (2.24)$$

Here, $z = x + iy$ takes values on the upper half plane, and $T(x)$ and $\tilde{T}(x)$ are arbitrary functions. The corresponding spin connection one-forms are

$$\omega = -\left(\frac{1}{y} + yT(x)\right)dx, \qquad \omega = -\left(\frac{1}{y} + y\tilde{T}(x)\right)dx. \qquad (2.25)$$

---

[6] While (2.23) is a correct correct local description of the phase space of (2+1)D gravity, it was argued in [17, 18] that invariance under large diffeomorphisms requires a further quotient by the diagonal action of the mapping class group $M$, which leads to $\frac{\mathcal{T}(g,b)\times\mathcal{T}(g,b)}{M}$. Such global aspects will however not play a role in the semiclassical considerations in this work.

The vector fields $V$ and $\tilde{V}$ which determine the gauge choice (2.11) are taken to be

$$V = -\tilde{V} = -\partial_z - \partial_{\bar{z}} = -\partial_x. \tag{2.26}$$

The resulting time evolution equation (2.19) is simply solved by replacing $x \to x_+ = x + t$ in $e$ and $x \to x_- = x - t$ in $\tilde{e}$. One checks that the (2+1)D metric (2.20) is indeed in of Banados form [24], i.e.

$$ds^2 = \frac{dy^2 + dx_+ dx_-}{y^2} - T(x_+)dx_+^2 - \tilde{T}(x_-)dx_-^2 + y^2 T(x_+)\tilde{T}(x_-)dx_+ dx_-. \tag{2.27}$$

The functions $-kT(x_+)$ and $-k\tilde{T}(x_-)$ are identified with the components of boundary stress tensor [25, 26].

**Residual gauge fixing: conformal gauge**

Reverting to the general case where $\Sigma_0$ has arbitrary topology, we will in this paper use the residual gauge freedom (2.14) to bring the auxiliary metrics (2.18) in the conformal gauge. Let us first introduce the following explicit parametrization of the zweibeins:

$$e = e^{-(\Phi + i\lambda)}(dz + \mu d\bar{z}), \qquad \tilde{e} = e^{-(\tilde{\Phi} + i\tilde{\lambda})}(dz + \tilde{\mu} d\bar{z}), \tag{2.28}$$

where $\Phi, \lambda$ are real fields and $\mu$ is complex (and similarly for their tilded counterparts). We will also restrict to

$$|\mu| < 1, \qquad |\tilde{\mu}| < 1 \tag{2.29}$$

so that the zweibein $(e, \bar{e})$ is invertible. We should keep in mind that the above is a parametrization for three-dimensional gauge potentials and that the fields $\Phi, \lambda, \mu, \omega$ (and their tilded counterparts) depend on all three coordinates $(t, z, \bar{z})$.

Let us now display the equations of motion in this parametrization, starting with the constraint equations (2.12). These reduce to

$$\omega_z = -\partial_z \lambda - i(1 - |\mu|^2)^{-1}\left((1 + |\mu|^2)\partial_z \Phi + \partial_{\bar{z}}\bar{\mu} - \bar{\mu}(\partial_z \mu + 2\partial_{\bar{z}}\Phi)\right) \tag{2.30}$$

$$e^{-2\Phi} = \frac{\operatorname{Im}\partial_{\bar{z}}\omega_z}{1 - |\mu|^2} \tag{2.31}$$

The residual gauge freedom (2.14) can be used to bring both auxiliary metrics $ds_2^2$ and $d\tilde{s}_2^2$ in conformal gauge on the $t = 0$ $\Sigma_0$, i.e.

$$\mu = \lambda = 0, \qquad \tilde{\mu} = \tilde{\lambda} = 0, \qquad \text{at } t = 0 \tag{2.32}$$

The constraint equations (2.12) at $t = 0$ reduce to $\omega_z = -i\partial_z \Phi$, $\tilde{\omega}_z = -i\partial_z \tilde{\Phi}$ and Liouville's equation for $\Phi$ and $\tilde{\Phi}$:

$$\partial_z \partial_{\bar{z}}\Phi + e^{-2\Phi} = 0, \qquad \partial_z \partial_{\bar{z}}\tilde{\Phi} + e^{-2\tilde{\Phi}} = 0, \qquad \text{at } t = 0. \tag{2.33}$$

On $\Sigma_0$, the spatial gauge field $\hat{A}$ takes the form

$$\hat{A} = \hat{A}[\Phi] := e^{-\Phi}dz L_1 - e^{-\Phi}d\bar{z} L_{-1} + (\partial_z \Phi dz - \partial_{\bar{z}}\Phi d\bar{z})L_0, \qquad \text{at } t = 0. \tag{2.34}$$

In this we recognize the standard form for the Lax connection for the Liouville equation (see e.g. [27]). When particle sources are coupled to gravity, (2.33) will be modified by delta-function sources as we will discuss in detail in Section 2.4. When the spacetime has a conformal boundary, these should be supplemented with appropriate AdS boundary conditions which we will discuss in Section 3 below.

From the Liouville solutions (3.6) we can construct two holomorphic stress tensors on the initial value surface as follows:

$$T(z) = -(\partial_z \Phi^2 + \partial_z^2 \Phi) \qquad \tilde{T}(z) = -(\partial_z \tilde{\Phi}^2 + \partial_z^2 \tilde{\Phi}) \qquad \text{at } t = 0. \tag{2.35}$$

These will play an important role in the rest of this work.

Now let us consider the dynamical equations (2.19) which determine the time dependence of the fields. They lead to the coupled first order equations

$$\partial_t(\Phi + i\lambda) = -\left((V^z \partial_z + V^{\bar{z}} \partial_{\bar{z}})(\Phi + i\lambda) + V^{\bar{z}} \partial_z \mu\right) + \partial_z(V^z + \mu V^{\bar{z}})$$
$$\partial_t \mu = -\left(\partial_{\bar{z}} - \mu \partial_z + \partial_z \mu\right)(V^z + \mu V^{\bar{z}}). \tag{2.36}$$

These should be solved with (2.32) and the solutions to (2.33) as initial conditions. We note from (2.36) that in general the fields $\mu$ and $\lambda$ will not stay zero at later times.

### Initial data for 3D gravity

The constraint- and dynamical equations (2.12,2.13) guarantee that the (2+1)D metric (2.20) satisfies Einsteins equations with negative cosmological constant,

$$R_{\mu\nu} = -2g_{\mu\nu}. \tag{2.37}$$

In our parametrization (2.15,2.28) and gauge choice (2.32), the induced metric on the initial value surface $\Sigma_0$ takes the simple form

$$ds_{\Sigma_0}^2 = \left(e^{-\Phi} + e^{-\tilde{\Phi}}\right)^2 dz d\bar{z} - \left[\Im m\left(\partial_z(\Phi - \tilde{\Phi}) dz\right)\right]^2 \tag{2.38}$$

We note that this metric is form-invariant under conformal transformations off $\Sigma_0$ with the Liouville fields transforming in the standard way,

$$z \to z' = f(z), \qquad \Phi \to \Phi' = \Phi + \frac{1}{2}\ln(f'\bar{f}'), \qquad \tilde{\Phi} \to \tilde{\Phi}' = \tilde{\Phi} + \frac{1}{2}\ln(f'\bar{f}'). \tag{2.39}$$

The Liouville (2.33) and time evolution equations (2.36) ensure that the 2D metric (2.38) and the extrinsic curvature $K$ (whose form is rather complicated) on the initial slice satisfy the initial value constraints of (2+1)D gravity:

$$R^{(2)} + (K_\mu^\mu)^2 - K_{\mu\nu}K^{\mu\nu} + 2 = 0$$
$$D_\mu K_\nu^\mu - D_\nu K_\mu^\mu = 0, \tag{2.40}$$

where $D_\mu$ is the covariant derivative with respect to the metric (2.38) on $\Sigma_0$. In classical general relativity, the phase space consists of initial data satisfying (2.40). In our formalism this data is repackaged in the form of two fields solving Liouville's equation. As we shall presently see, this becomes especially advantageous when including point particle matter sources. The nontrivial problem of finding consistent initial data then reduces to solving the Liouville equation with delta-function sources, for which there exists a well-developed mathematical physics machinery.

## 2.4 Spinning particle sources and backreaction

We now consider gravity coupled to massive, spinning matter fields in a limit in which these fields behave like heavy point particles backreacting on the geometry. Let us say a little more about this point particle limit. A quantum field of mass $m$ and helicity $s$ propagates a lowest weight representation of $sl(2, \mathbb{R}) \times sl(2, \mathbb{R})$ built on a primary of weight $(h, \tilde{h})$, where (see e.g. [28])

$$m^2 = (h + \tilde{h})(h + \tilde{h} - 2), \qquad s = h - \tilde{h} \tag{2.41}$$

High-energy quanta are expected to behave as point particles. Since the AdS energy is given by $L_0 + \tilde{L}_0$, the point particle limit is $h + \tilde{h} \gg 2$ and the relations (2.41) simplify to

$$m = h + \tilde{h}, \qquad s = h - \tilde{h}. \tag{2.42}$$

The description of point particle sources in Chern-Simons variables goes back to [29], see [30],[31] for more recent discussions. They are described by a classical source action which, upon quantization, leads to a Wilson line in the appropriate unitary irreducible representation of the gauge group. A point particle with mass $m$ and helicity $s$ is described by the following worldline action

$$S_p[U, P, \lambda; A; h] = \int_C d\tau \left[ \text{tr}(P D_\tau U U^{-1}) + \lambda \left( \text{tr} P^2 + 2h^2 \right) \right] \tag{2.43}$$

$$D_\tau U \equiv \frac{dU}{d\tau} + A_\tau U, \tag{2.44}$$

where $A_\tau$ is defined as the component of the connection parallel to $C$:

$$A_\tau := A_\mu(x(\tau)) \frac{dx^\mu}{d\tau} \tag{2.45}$$

A similar action describes the worldline coupling to $\tilde{A}$. Here, $\tau$ parametrizes the worldline $C$ and the dynamical variables $U$ and $P$ are elements of the $SL(2, \mathbb{R})$ group and $sl(2, \mathbb{R})$ Lie algebra respectively. The action is invariant under worldline reparametrizations, and the combined Chern-Simons and source actions remain invariant under $SL(2, \mathbb{R})$ gauge transformations $A \to \Lambda^{-1}(A + d)\Lambda$ provided that the worldline fields transform as

$$U \to \Lambda^{-1} U, \qquad P \to \Lambda^{-1} P \Lambda. \tag{2.46}$$

The gauge invariance of the total action means in particular that it is independent of the shape of the curve $C$; it is therefore not necessary to vary this action with respect to the worldline $C$.

The equations of motion following from varying the particle action with respect to the worldline variables $U, P, \lambda$ reduce to

$$\frac{dP}{d\tau} + [A_\tau, P] = 0, \qquad\qquad \frac{d\tilde{P}}{d\tau} + [\tilde{A}_\tau, \tilde{P}] = 0 \tag{2.47}$$

$$\text{tr} P^2 = -2h^2, \qquad\qquad \text{tr} \tilde{P}^2 = -2\tilde{h}^2 \tag{2.48}$$

$$\frac{dU}{d\tau} U^{-1} + A_\tau + 2\lambda P = 0, \qquad\qquad \frac{d\tilde{U}}{d\tau} \tilde{U}^{-1} + \tilde{A}_\tau + 2\tilde{\lambda}\tilde{P} = 0 \tag{2.49}$$

The equations for the momenta $P$ and $\tilde{P}$ in the first line can be shown [31] to be equivalent to the Mathisson-Papapetrou-Dixon (MPD) equations [32–34] governing the motion of spinning point particles in general relativity, as we review in Appendix A. In the metric formulation these express conservation of momentum and angular momentum along the wordline, and generalize the geodesic motion required for consistent coupling of nonspinning particles to gravity. As explained in Appendix A, the (2+1)D MPD equations admit solutions describing standard geodesic motion, but also allow for more general types of motion in the spinning case. The subclass of geodesic solutions corresponds to (see (A.13))

$$[P, \tilde{P}] = [E_\tau, P] = [E_\tau, \tilde{P}] = 0. \tag{2.50}$$

Varying the total action with respect to $A$ yields the following source term for the field strength

$$F_{\mu\nu} = -\frac{\pi}{k} \epsilon_{\mu\nu\rho} \int_C d\tau P \frac{dx^\rho}{d\tau} \delta^{(3)}(x - x(\tau)) \tag{2.51}$$

We see that, for a particle to backreact on the geometry, either $h$ or $\tilde{h}$ should grow linearly with $k$ in the semiclassical limit of large $k$. For later convenience we define the ratios

$$\alpha = \frac{h}{k}, \qquad \tilde{\alpha} = \frac{\tilde{h}}{k}. \tag{2.52}$$

It will also be useful in what follows to distinguish some special cases. We will call the particle 'chiral' if $\alpha$ stays finite but $\tilde{\alpha} \to 0$ in the large $k$ limit, and 'antichiral' if the converse is true. The particle is of 'generic' type if $\alpha\tilde{\alpha} \neq 0$. We will see in Section 3.6 that the case where all the particles involved are either of the chiral or anti-chiral type (i.e. $\alpha\tilde{\alpha} = 0$) is special in that the dynamics simplifies significantly.

Let us discuss the effect of the source terms in (2.51) and in the analogous equation for $\tilde{F}_{\mu\nu}$ in more detail. We will restrict attention to curves $C$ for which $t(\tau)$ is a monotonic function,

$$\frac{dt}{d\tau} > 0. \tag{2.53}$$

We can then choose the parameter of the curve to coincide with the time coordinate $t$,

$$x^\mu(\tau) = (\tau, z(\tau), \bar{z}(\tau)). \tag{2.54}$$

On general grounds, we expect that the introduction of the sources should modify the constraint equation (and hence the phase space), but not the dynamical equations. For a generic source with $\alpha\tilde{\alpha} \neq 0$ this is compatible with our gauge choice (2.11) if we choose the the vectors $V$ and $\tilde{V}$ to coincide, on the particle worldline, with the spatial velocity

$$V(x(t)) = \tilde{V}(x(t)) = \dot{z}\partial_z + \dot{\bar{z}}\partial_{\bar{z}}, \qquad \text{generic } (\alpha\tilde{\alpha} \neq 0). \tag{2.55}$$

Indeed, it is straightforward to see that the source term in (2.51) then does not modify the dynamic equation (2.13). For an (anti-)chiral particle we need only impose that $V$ (resp. $\tilde{V}$) become equal to the spatial velocity:

$$\begin{aligned}
V(x(t)) &= \dot{z}\partial_z + \dot{\bar{z}}\partial_{\bar{z}}, \qquad \text{chiral } (\tilde{\alpha} = 0) \\
\tilde{V}(x(t)) &= \dot{z}\partial_z + \dot{\bar{z}}\partial_{\bar{z}}, \qquad \text{antichiral } (\alpha = 0).
\end{aligned} \tag{2.56}$$

Note that these imply the vanishing of components of the gauge fields parallel to the worldline:

$$
\begin{aligned}
A_\tau = \tilde{A}_\tau = 0 & \qquad \text{generic } (\alpha\tilde{\alpha} \neq 0) \\
A_\tau = 0 & \qquad \text{chiral } (\tilde{\alpha} = 0) \\
\tilde{A}_\tau = 0 & \qquad \text{antichiral } (\alpha = 0).
\end{aligned}
\tag{2.57}
$$

Consequently the equations (2.47) impose that $P$ and $\tilde{P}$ are constant. The equation (2.48) can be solved as[7]

$$
P = -2hiL_0, \qquad \tilde{P} = -2i\tilde{h}L_0.
\tag{2.58}
$$

The remaining equation (2.49) is solved by e.g. taking

$$
\lambda = \tilde{\lambda} = 0, \qquad U = \tilde{U} = 1.
\tag{2.59}
$$

Substituting (2.58) into (2.51) we find that the constraint equations (2.33) on the $t = 0$ slice are modified to[8]

$$
\partial_z\partial_{\bar{z}}\Phi + e^{-2\Phi} = \pi\sum_{i=1}^{n}\alpha_i\delta^{(2)}(z - z_i), \qquad \partial_z\partial_{\bar{z}}\tilde{\Phi} + e^{-2\Phi} = \pi\sum_{i=1}^{n}\tilde{\alpha}_i\delta^{(2)}(z - z_i).
\tag{2.60}
$$

Here, we have performed the straightforward generalization to include $n$ particle sources labelled by $i$. Each delta-function source term creates a deficit angle of $2\pi\alpha_i$ at the point $z_i$ in the 2D metric $ds^2$. We will restrict to $0 \leq \alpha_i < 1$ as appropriate for particle sources below the BTZ black hole treshold which corresponds to $\alpha = 1$.

It may seem surprising that the wordlines $C_i$ were largely arbitrary (except for the assumption (2.53)), and yet they end up satisfying the MPD equations. The reason for this is that the vectors $V$ and $\tilde{V}$, which depend on the $C_i$ through (2.55,2.56), enter in the (2+1)D metric (2.20) so as to ensure that the MPD equations are obeyed. We are working in a 'Kantian' formulation where the object (the metric) directs itself towards our knowledge (of the worldlines) and not vice versa.

It is enlightening to check whether the particles in the backreacted solutions move on geodesics. Recall that this requires (2.50) to hold. In the non-chiral case, this is so due to the fact that (2.57) implies that the (2+1)D vielbein degenerates on the wordline:

$$
E_\tau = \frac{1}{2}(A_\tau - \tilde{A}_\tau) = 0 \qquad \text{(generic)}.
\tag{2.61}
$$

The coordinate singularity signalled by the degeneration of the vielbein here reflects the conical curvature singularity on the worldline [14, 15]. For chiral particles however, the vielbein need not be degenerate on the worldline as $E_\tau = \tilde{A}_\tau/2$. They follow geodesics if, at $t = 0$,

$$
\tilde{A}_\tau \sim L_0.
\tag{2.62}
$$

_____________________

[7]The factors $i$ are a consequence of writing the equations in the $su(1,1)$ basis, see Footnote 4.

[8]In our conventions, $\epsilon_{tz\bar{z}} = i$ and the (2+1)D delta function is normalized such that $1 = \int dt d^2z \delta^{(3)}(x) = 2\int dt dx dy \delta^{(3)}(x)$.

If not, the chiral particle follows a more general trajectory which solves the MPD equations.

As the above analysis indicates, the phase space of (2+1) gravity in the presence of particle sources, is (locally) the product of Teichmüller spaces of punctured Riemann surfaces,

$$\mathcal{T}(g, b, n) \times \mathcal{T}(g, b, \tilde{n}), \tag{2.63}$$

where $n$ ($\tilde{n}$) is the number of particles with nonvanishing quantum number $\alpha$ (resp. $\tilde{\alpha}$). For example, for $g = 0, b = 1, n = 1$, the Teichmüller space $\mathcal{T}(0, 1, 1)$ can be identified, as a symplectic manifold, with the Virasoro coadjoint orbit $\mathrm{Diff}\, S^1/U(1)$ [16],[21]. Upon quantization, one obtains a generic primary representation of the Virasoro algebra. It would be of interest to have a better mathematical understanding of the Teichmüller spaces with $n > 1$ and $b \geq 1$.

# 3 Asymptotically Anti-de Sitter spacetimes

So far we did not specify the topology of the spatial slice $\Sigma$. In this and the following section, we will focus on the case where $\Sigma$ has a single asymptotic boundary, which is the standard setting for the AdS/CFT correspondence. In particular we will be describing multi-particle excitations on a global Anti-de Sitter background. We will comment on the case where $\Sigma$ has spherical topology in Section 4.

When there is a asymptotic boundary, the initial value surface $\Sigma_0$ has the topology of an open disk, possibly punctured by particle sources. The Liouville fields $\Phi$ and $\tilde{\Phi}$ describe hyperbolic metrics on this surface. We also want to impose asymptotically AdS boundary conditions in the standard sense of Brown and Henneaux [12]. We will see that this reduces, in our parametrization, to pseudosphere or Zamolodchikov-Zamolodchikov (ZZ) boundary conditions [6] on the Liouville fields $\Phi$ and $\tilde{\Phi}$. Using an appropriate doubling trick we will extend all quantities to the Riemann sphere, and in particular find a connection to spherical conformal blocks.

## 3.1 AdS asymptotics and the pseudosphere

We can model $\Sigma_0$ as the complex upper half plane parametrized by a complex coordinate $z$ with

$$\mathrm{Im}\, z \geq 0. \tag{3.1}$$

The AdS conformal boundary corresponds to the real line, compactified by adding the point at infinity. As we will see, this coordinate system will describe physics in the Poincaré patch. This will be sufficient for most purposes and leads to simple formulas. The extension to a global coordinate system will be discussed at the end of this subsection.

In the $(t, z, \bar{z})$ coordinate system the standard boundary conditions [24] on the Chern-Simons gauge potentials read

$$A_- := A_t - A_z - A_{\bar{z}} = 0 \tag{3.2}$$

$$\tilde{A}_+ := A_t + A_z + A_{\bar{z}} = 0 \qquad \text{at } \mathrm{Im}\, z = 0, \tag{3.3}$$

In addition, we will impose suitable fall-off conditions [24] to ensure that the resulting spacetime is asymptotically AdS.

The boundary conditions (3.3) are consistent with our gauge choice (2.11) provided that the vector fields $V, \tilde{V}$ behave near the boundary as:

$$V \to -\partial_z - \partial_{\bar{z}}, \qquad \tilde{V} \to \partial_z + \partial_{\bar{z}} \qquad \text{as Im } z \to 0. \tag{3.4}$$

As described in the previous section, in the presence of particle sources, we also require the vector fields $V, \tilde{V}$ to coincide, on the particle worldlines in the interior, with their spatial velocities (2.55, 2.56). Let us however first discuss the situation without sources. We can then simply take

$$V = -\tilde{V} = -\partial_z - \partial_{\bar{z}} \tag{3.5}$$

throughout the spacetime. The constraint- and dynamical equations (2.33, 2.36) then reduce to

$$0 = \partial_z \partial_{\bar{z}} \Phi + e^{-2\Phi}, \qquad\qquad 0 = \partial_z \partial_{\bar{z}} \tilde{\Phi} + e^{-2\tilde{\Phi}}, \tag{3.6}$$

$$\partial_t \Phi = (\partial_z + \partial_{\bar{z}})\Phi, \qquad\qquad \partial_t \tilde{\Phi} = -(\partial_z + \partial_{\bar{z}})\tilde{\Phi} \tag{3.7}$$

$$\mu = \lambda = 0, \qquad\qquad \tilde{\mu} = \tilde{\lambda} = 0 \tag{3.8}$$

In other words, the auxiliary 2D metrics (2.18) remain in conformal gauge at all times, and the time dependence is such that $\Phi$ and $\tilde{\Phi}$ depend only on the combinations $z_+ := z + t$ and $z_- := z - t$ respectively. Their real parts $x_\pm = \text{Re}(z) \pm t$ will turn out to play the role of light-cone coordinates on the boundary.

The 2+1 dimensional metric (2.20) takes the form

$$ds^2 = \left| e^{-\Phi(z_+, \bar{z}_+)} dz_+ + e^{-\tilde{\Phi}(z_-, \bar{z}_-)} dz_- \right|^2 - \left[ \text{Im} \left( \partial_{z_+} \Phi(z_+, \bar{z}_+) dz_+ - \partial_{z_-} \tilde{\Phi}(z_-, \bar{z}_-) dz_- \right) \right]^2 \tag{3.9}$$

One can check that the (2+1)D Einstein equations $R_{\mu\nu} + 2g_{\mu\nu} = 0$ indeed reduce to (3.6).

An elementary solution to (3.7) is to take $\Phi$ and $\tilde{\Phi}$ to coincide at $t = 0$ and take the form

$$\Phi = \tilde{\Phi} = \ln 2\text{Im } z, \qquad \text{at } t = 0. \tag{3.10}$$

We note that $\Phi, \tilde{\Phi}$ diverge on the asymptotic boundary, where $z$ is real. The corresponding auxiliary 2D metrics (2.18) describe the hyperbolic metric on the upper half plane:

$$ds_2^2 = d\tilde{s}_2^2 = \frac{dz d\bar{z}}{4(\text{Im} z)^2}. \tag{3.11}$$

The Liouville stress tensors (2.35) associated to (3.10) vanish,

$$T(z) = \tilde{T}(z) = 0, \tag{3.12}$$

and the (2+1)D metric (3.9) is simply AdS$_3$ in Poincaré coordinates:

$$ds^2 = \frac{-dt^2 + dz d\bar{z}}{(\text{Im} z)^2}. \tag{3.13}$$

More generally we will consider Liouville solutions which approach (3.10) near the real axis,

$$\Phi = \ln(2\text{Im}\, z) + \mathcal{O}(1), \qquad \tilde{\Phi} = \ln(2\text{Im}\, z) + \mathcal{O}(1). \tag{3.14}$$

In the literature on boundary Liouville theory, these are known as Zamolodchikov-Zamolodchikov (ZZ) or pseudosphere boundary conditions [6].

We will now show that the boundary conditions (3.14) give rise to $(2+1)$D metrics obeying the Brown-Henneaux falloff conditions. As explained in [7], the Liouville equation determines the first subleading correction to (3.14) in terms of the stress tensor on the boundary:

$$\Phi = \ln(2\text{Im}\, z) + \frac{2}{3}(\text{Im}\, z)^2 T_{|z=\bar{z}} + \mathcal{O}(\text{Im}\, z)^4 \tag{3.15}$$

$$\tilde{\Phi} = \ln(2\text{Im}\, z) + \frac{2}{3}(\text{Im}\, z)^2 \tilde{T}_{|z=\bar{z}} + \mathcal{O}(\text{Im}\, z)^4 \tag{3.16}$$

The higher order terms in in this expansion are determined by the value of the stress tensor on the real line. The expansion (3.16) and the reality of $\Phi, \tilde{\Phi}$ imply in particular that the value of the stress tensor on the real axis is real:

$$gT(x) = \bar{T}(x), \qquad \tilde{T}(x) = \bar{\tilde{T}}(x), \qquad \text{for } x \in \mathbb{R} \tag{3.17}$$

To bring the metric in Fefferman-Graham form, we adopt following parametrization

$$z = x + i\left(y - \frac{1}{6}\left(T(x_+) + \tilde{T}(x_-)\right)y^3 + \mathcal{O}(y^5)\right) \tag{3.18}$$

and find

$$ds^2 = \frac{dy^2 + dx_+ dx_-}{y^2} - T(x_+)dx_+^2 - \tilde{T}(x_-)dx_-^2 + \mathcal{O}(y^2). \tag{3.19}$$

In particular, the real functions $-kT(x_+)$ and $-k\tilde{T}(x_-)$ coincide with the components of the boundary stress tensor [25, 26] of the asymptotically AdS spacetime. We note from (3.18) that the imaginary part of $z$ tends to the Fefferman-Graham radial coordinate $y$ near the boundary. The subleading correction in (3.18) is necessary to remove an unwanted term of order one in $g_{yy}$.

When particle sources are present, we can still arrange for $V$ and $\tilde{V}$ to take the form (3.5) in a neighborhood of the asymptotic boundary. Therefore the asymptotic form of the metric (3.19) and the relation between the Liouville and boundary stress tensors continues to hold in this case. Near the delta-function sources $z_i$ in (2.60), the Liouville field has the asymptotics

$$\Phi \overset{z \to z_i}{\sim} \alpha_j \ln|z - z_i| + \mathcal{O}(1), \qquad j = 1, \ldots, n. \tag{3.20}$$

Correspondingly, the stress tensor $T(z)$ has a second order pole

$$T(z) \overset{z \to z_i}{\sim} \frac{\epsilon_j}{(z - z_i)^2} + \mathcal{O}\left((z - z_i)^{-1}\right), \qquad i = 1, \ldots, n, \tag{3.21}$$

where we defined

$$\epsilon_i = \frac{\alpha_i}{2}\left(1 - \frac{\alpha_i}{2}\right). \tag{3.22}$$

The stress tensor $T(z)$ is therefore meromorphic on the upper half plane. Similar properties hold for $\tilde{\Phi}$ and $\tilde{T}(z)$.

**Other useful coordinate systems**

It is often convenient to model $\Sigma_0$ as a punctured disk instead instead of the above upper half plane model; this has the advantage that the full spacetime accessible as a bounded domain. For this purpose we make a conformal transformation on $\Sigma_0$:

$$z = i\frac{1-w}{1-w}. \tag{3.23}$$

The new coordinate runs over the unit disk $|w| \leq 1$, and the AdS vacuum solution corresponds in this frame to

$$\Phi = \tilde{\Phi} = \ln(1 - |w|^2), \qquad \text{at } t = 0. \tag{3.24}$$

A natural choice for the vector fields $V, \tilde{V}$ specifying the Chern-Simons gauge choice is

$$V = -iw\partial_w + i\bar{w}\partial_{\bar{w}}, \qquad \tilde{V} = iw\partial_w - i\bar{w}\partial_{\bar{w}}. \tag{3.25}$$

In Appendix B we show that the corresponding (2+1)D metric is of the form (3.9), with $z_\pm$ replaced by the combinations $w_\pm$ defined as

$$w_+ = we^{it}, \qquad w_- = we^{-it}. \tag{3.26}$$

To conclude, we explain how to set up the equations in global coordinates. To this end we make a further conformal transformation on $\Sigma_0$ to a coordinate $u$ defined on the semi-infinite cylinder,

$$w = e^{iu}, \qquad \text{Im } u \geq 0, \qquad u \sim u + 2\pi. \tag{3.27}$$

In the gauge determined by vector fields $V, \tilde{V}$ given by

$$V = -\partial_u - \partial_{\bar{u}}, \qquad \tilde{V} = \partial_u + \partial_{\bar{u}}, \tag{3.28}$$

the 2+1 dimensional metric again takes the form (3.9) with $z_\pm$ replaced by $u_\pm$ defined as

$$u_+ = u + t, \qquad u_- = u - t. \tag{3.29}$$

Conformally mapping the AdS solution (3.24) gives

$$\Phi = \tilde{\Phi} = \ln 2 \sinh \text{Im } u \qquad \text{at } t = 0. \tag{3.30}$$

The stress tensors in this frame are $T(u) = \tilde{T}(\bar{u}) = \frac{1}{4}$. The (2+1)D metric is simply AdS$_3$ in global coordinates:

$$ds^2 = d\rho^2 - \cosh^2 \rho \, dt^2 + \sinh^2 \rho \, d\phi^2. \tag{3.31}$$

where we parametrized $u$ as

$$u = \phi - i \ln \tanh \frac{\rho}{2}. \tag{3.32}$$

**Pseudosphere doubling trick**

As is usually the case for CFTs defined on a bounded domain of the complex plane with certain boundary conditions, it is possible to apply a 'doubling trick' which extends the theory to the full complex plane in such a way as to automatically satisfy the boundary conditions. For the case of Liouville theory with ZZ boundary conditions this was worked out in [7].

As we argued above, the Liouville stress tensor $T(z)$ is a meromorphic function on the upper half plane which, due to the ZZ boundary conditions, takes on real values on the real line (3.17) . It can therefore be extended a meromorphic function on the extended complex plane $\overline{\mathbb{C}}$ satisfying

$$T(z) = \bar{T}(z). \tag{3.33}$$

We recall the general form of the Liouville solution,

$$e^{-2\Phi} = \frac{|f'|^2}{4(\operatorname{Im} f)^2} \tag{3.34}$$

where $f$ satisfies

$$S(f, z) = 2T(z), \tag{3.35}$$

and $S(f, z)$ denotes the Schwarzian derivative.

The reflection condition (3.33) translates into a similar reflection condition on $f$,

$$f(z) = \bar{f}(z). \tag{3.36}$$

Upon extending $f$ to the complex plane, we obtain through (3.34) a Liouville field encoding two joined 'hemispheres' of a pseudosphere, with singularities in image points, as sketched in Figure 4(a). In Appendix B we give more details on the doubling trick in the unit disk conformal frame.

The function $f(z)$ in (3.34) is multivalued with branch points at each of the source locations. In Section 3.3 below we will give more details on how $f$ is determined from the solution of a certain monodromy problem.

## 3.2 Hartle-Hawking wavefunction and Liouville action

In this section we will consider a Chern-Simons path integral, which is analogous to the Hartle-Hawking [35] path integral over metrics, and prepares a multi-particle state on the disk $\Sigma_0$ at $t = 0$. The particles have worldlines which start from the conformal boundary in the past and terminate on $\Sigma_0$, as in the lower half of Figure 1(b). This can also be depicted as in Figure 2: the manifold $X$ on which we compute the Chern-Simons path integral is a filled-in pseudosphere, where $\Sigma_0$ is the boundary of the northern 'hemisphere'. The particle worldlines emerge from the southern hemisphere and end up at $\Sigma_0$. We can view them as connecting image points on the pseudosphere through the bulk. We consider $n$ particles with quantum numbers $\alpha_i$ located at positions $z_i$ in the upper half plane model for $\Sigma_0$. Their images on the southern hemisphere are located at $\bar{z}_i$.

The wavefunction of interest should be a functional of the canonical coordinate fields on the boundary of $X$, which are to be held fixed in the path integral. Depending on the

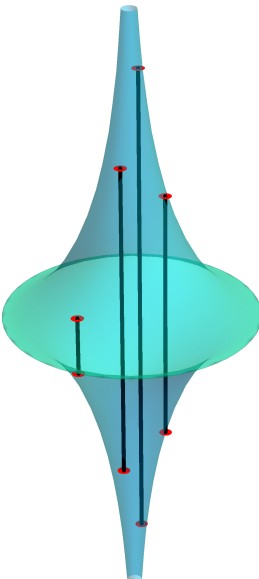

**Figure 2**. The lower half of Figure 1(b) can be redrawn as a filled-in pseudosphere with particle worldlines connecting image points through the bulk.

choice of canonical coordinate one obtains wavefunctions in different polarizations. Here we will choose a holomorphic polarization[9], where $A_z$ and $\tilde{A}_{\bar{z}}$ are treated as coordinates and $A_{\bar{z}}$ and $\tilde{A}_{\bar{z}}$ as their conjugate momenta. In order to have a consistent variational principle under the these boundary conditions, it straightforward to see that one has to add a boundary term to the action which can be taken to be [36]

$$S_{bdy}[A] = \frac{k}{4\pi}\mathrm{tr}\int_{\mathbb{C}} dz \wedge d\bar{z}\, A_z A_{\bar{z}}. \tag{3.37}$$

We note this boundary term is actually purely imaginary. The fact that we require a non-real boundary term is a consequence of choosing a non-real polarization.

The Hartle-Hawking-like path integral of interest[10] factorizes as

$$\Psi_{HH}[A_z, \tilde{A}_{\bar{z}}; z_i, \bar{z}_i; \alpha_i, \tilde{\alpha}_i] = \Psi[A_z; z_i, \bar{z}_i; \alpha_i]\tilde{\Psi}[\tilde{A}_{\bar{z}}; z_i, \bar{z}_i; \tilde{\alpha}_i], \tag{3.38}$$

where

$$\Psi[A_z; z_i, \bar{z}_i; \alpha_i] = \int_X [DADU_i DP_i D\lambda_i]\big|_{A_z} e^{i\left(S_{CS}[A]+S_{bdy}[A]+\sum_i S_p[U_i,P_i,\lambda_i;A;\alpha_i]\right)}, \tag{3.39}$$

and $\tilde{\Psi}$ is defined analogously. We recall that $S_p$ is the worldline action of the particles given in (2.44).

In the standard approach to quantizing Chern-Simons theory with compact gauge groups on compact manifolds, a similar functional gives a path integral representation of

---

[9]Another common choice [16], where $(e, \omega_z)$ are treated as coordinates and $(\bar{e}, \omega_{\bar{z}})$ as momenta, leads to a different boundary term $S_{bdy} = -\frac{k}{8\pi}\int_{\mathbb{C}} dz \wedge d\bar{z}(|\omega_z|^2 - 2e^{-2\Phi}(1-|\mu|^2))$, which however coincides with (3.37) in conformal gauge (see (3.46) below).

[10]In principle we could have considered a more general functional which also depends on the boundary values of the worldline variable $U$, but we will specialize here to the boundary value $U = 1$.

Kac-Moody blocks [37]. For noncompact gauge groups and the extension to manifolds $\Sigma_0$ with asymptotic boundary, the exact evaluation of path integrals of the form (3.39) is progressively less straightforward, see [38] for a recent discussion. In this work, we will focus on a more pedestrian goal and consider the large $k$ limit where (3.39) is dominated by a classical saddle point, which we will then relate to the large $k$ limit of an appropriate CFT block. At large $k$, the wavefunction $\Psi[A_z; z_i, \bar{z}_i; h_i]$ is well approximated by the classical action

$$\Psi[A_z; z_i, \bar{z}_i; \alpha_i] \overset{k \to \infty}{\sim} e^{iS_{cl}^{tot}}, \tag{3.40}$$

where the total action $S^{tot} = S_{CS} + S_{bdy} + \sum_i S_{p,i}$ includes the boundary and source terms. It is to be evaluated on the classical solution which takes on the specified value $A_z$ on the boundary.

We will first show that the total on-shell action receives contributions only from the boundary term $S_{bdy}$ in (3.37). Recalling that, in our chosen gauge, the component $A_\tau$ parallel to the worldline vanishes, and using the solution (2.58,2.59) for the worldline fields, one sees that the source action $S_p$ vanishes. The bulk Chern-Simons action can be rewritten as

$$S_{CS}[A] = \frac{k}{4\pi} \text{tr} \int_X \left( A \wedge F - \frac{1}{3} A \wedge A \wedge A \right). \tag{3.41}$$

The second term vanishes since $\det A_\mu^a = 0$ in our gauge (2.11), while the first term vanishes upon using (2.51) and (2.57). Therefore only the boundary term contributes to the total action and gives

$$\Psi[A_z; z_i, \bar{z}_i; \alpha_i] \sim \exp \frac{ik}{4\pi} \text{tr} \int_\mathbb{C} dz \wedge d\bar{z} A_z A_{\bar{z}}^{cl}. \tag{3.42}$$

Here, $A_{\bar{z}}^{cl}$ is a solution to

$$\partial_z A_{\bar{z}}^{cl} - \partial_{\bar{z}} A_z + [A_z, A_{\bar{z}}^{cl}] = -2\pi \sum_i \alpha_i \delta^{(2)}(z - z_i) L_0. \tag{3.43}$$

From this expression, we learn that the argument $A_z$ is cannot completely arbitrary: the wavefunction has support on those $A_z$ which arise as components of an almost everywhere flat connection[11]. In other words, $A_z$ must locally be of the form $A_z = G^{-1} \partial_z G$, and due to (3.43) we must have, locally and in the $sl(2, \mathbb{R})$ basis, $A_{\bar{z}}^{cl} = G^{-1} \partial_{\bar{z}} G$. In order to match up with our conformal gauge choice on $\Sigma_0$ we will further specialize the wavefunction to those $A_z$ which are of the form (see (2.34))

$$A_z = e^{-\Phi} L_1 + \partial_z \Phi L_0 \Rightarrow A_{\bar{z}}^{cl} = -e^{-\Phi} L_{-1} - \partial_{\bar{z}} \Phi L_0, \tag{3.44}$$

where $\Phi$ satisfies (2.60). The resulting specialized wavefunction then depends only on $z_i, \bar{z}_i; \alpha_i$ and we will write it as $\Psi[z_i, \bar{z}_i; \alpha_i]$ in what follows.

Summarizing, we have argued that

$$\Psi[z_i, \bar{z}_i; \alpha_i] \sim e^{iS_{bdy}'[\Phi]}, \tag{3.45}$$

---

[11]This is a well-known fact [39] which, from the form (2.7), can be seen to extend to the full quantum functional integral.

where

$$S'_{bdy}[\Phi] = -\frac{k}{8\pi} \int_{\overline{\mathbb{C}}} dz \wedge d\bar{z} \left( \partial_z \Phi \partial_{\bar{z}} \Phi - 2e^{-2\Phi} \right) \tag{3.46}$$

Our final result will contain two corrections to this expression. First of all, (3.46) is not finite due to divergent contributions from the punctures. To regularize it, we will replace the integration region by $\overline{\mathbb{C}}_\epsilon$, which is the extended complex plane $\bar{\mathbb{C}}$ with small discs of radius $\epsilon$ removed around each of the source insertions $z_j$ and their images $\bar{z}_j$, as well as small strips of width $\epsilon$ above and below the real axis. In order to render the action finite upon taking the limit $\epsilon \to 0$, we see from the behavior (3.20) near the punctures that we should add the following term to (3.46):

$$r_\epsilon = -2\pi \sum_{j=1}^n \alpha_j^2 \ln \epsilon. \tag{3.47}$$

Note that it is independent of the source locations $z_i$.

A second modification is due to the fact that (3.46) is not differentiable as a functional of $\Phi$, again due to the source terms. The need to insist on differentiability was stressed in a related context in [40]. The variation of the integral (3.46) picks up boundary terms in the domain $\overline{\mathbb{C}}_\epsilon$

$$\delta S'_{bdy}[\Phi] = -\frac{k}{8\pi} \lim_{\epsilon \to 0} \int_{\delta \overline{\mathbb{C}}_\epsilon} \delta \Phi \left( \partial_{\bar{z}} \Phi d\bar{z} - \partial_z \Phi dz \right) + \dots \tag{3.48}$$

This variation can be cancelled on fields which behave as (3.14,3.20) by adding contour integrals along the boundary components of $\overline{\mathbb{C}}_\epsilon$. Upon doing so we arrive at the following finite and differentiable boundary action

$$S^\epsilon_{bdy}[\Phi] = \frac{k}{8\pi} \left( \int_{\overline{\mathbb{C}}_\epsilon} dz \wedge d\bar{z} \left( |\partial_z \Phi|^2 - 2e^{-2\Phi} \right) + r_\epsilon \right. \tag{3.49}$$

$$-\sum_{j=1}^n \frac{\alpha_j}{2} \left( \oint_{C_j^\epsilon} \Phi \left( \frac{d\bar{z}}{\bar{z} - \bar{z}_j} - \frac{dz}{z - z_j} \right) + \oint_{\tilde{C}_j^\epsilon} \Phi \left( \frac{d\bar{z}}{\bar{z} - z_j} - \frac{dz}{z - \bar{z}_j} \right) \right) \tag{3.50}$$

$$\left. + \left( \int_{\mathbb{R}+i\epsilon} + \int_{\mathbb{R}-i\epsilon} \right) \Phi \frac{d\bar{z} + dz}{z - \bar{z}} \right) \tag{3.51}$$

Here, the $C_j$ and $\tilde{C}_j$ are circular contours of radius $\epsilon$ around the source in $z_j$ and its image in $\bar{z}_j$ respectively. All line integrals in the above expression are oriented as boundary components of $\overline{\mathbb{C}}_\epsilon$.

Our main observation is now that the functional (3.51) closely resembles the standard, regularized, Liouville action on the pseudosphere $S^\epsilon_L[\Phi]$, see [6], except for a wrong coefficient multiplying the Liouville potential $e^{-2\Phi}$. More precisely we have

$$iS^\epsilon_{bdy}[\Phi] = -\frac{k}{4} S^\epsilon_L[\Phi] + \frac{3ik}{8\pi} \int_{\overline{\mathbb{C}}_\epsilon} dz \wedge d\bar{z} e^{-2\Phi}. \tag{3.52}$$

Using the fact the source-free Liouville equation holds on $\overline{\mathbb{C}}_\epsilon$, making use of (3.14,3.20) and taking the $\epsilon \to 0$ limit, we find that

$$iS_{bdy}[\Phi] = -\frac{k}{4} S_L[\Phi_{cl}] - \frac{3k}{4} \sum_{j=1}^n \alpha_j. \tag{3.53}$$

Therefore we have established that the on-shell Chern-Simons action on $X$ is proportional to the Liouville action on the pseudosphere, up to a term independent of the insertion points $z_i$. We conclude that the classical approximation to the Hartle-Hawking wavefunction (3.38) is of the form

$$\Psi(z_i, \bar{z}_i, \alpha_i) \sim N(\alpha_i, \tilde{\alpha}_i) e^{-\frac{k}{4}(S_L[\Phi_{cl}(z_i, \bar{z}_i, \alpha_i)] + S_L[\tilde{\Phi}_{cl}(z_i, \bar{z}_i, \tilde{\alpha}_i)])}. \tag{3.54}$$

with $N(\alpha_i, \tilde{\alpha}_i)$ independent of the insertion points $z_i$.

### 3.3 Accessory parameters and pseudosphere Polyakov relations

The property (3.54) will allow us to relate the wavefunction to a classical Virasoro block. An important ingredient for this connection is a property of the on-shell Liouville action known as the Polyakov relation. We start by recalling, following [7, 41], the role of accessory parameters in the monodromy problem involved in solving the sourced Liouville equation (2.60) on the upper half plane with boundary conditions (3.14). We will then fill a gap in the literature and derive a Polyakov relation for Liouville theory on the pseudosphere, identifying the on-shell action as the generating function of the accessory parameters.

**Acessory parameters**

From (3.20) and regularity at infinity, it follows that the stress tensor of a solution to (2.60, 3.14) is a meromorphic function on $\overline{\mathbb{C}}$ of the form

$$T(z) = \sum_{i=1}^{n} \left( \frac{\epsilon_i}{(z - z_i)^2} + \frac{\epsilon_i}{(z - \bar{z}_i)^2} + \frac{c_i}{z - z_i} + \frac{\tilde{c}_i}{z - \bar{z}_i} \right) \tag{3.55}$$

The as yet undetermined coefficients $c_i$ and $\tilde{c}_i$ are called accessory parameters. They are constrained by the reflection condition (3.33) as well as the requirement of regularity as $z \to \infty$. This leads to $2n + 3$ real conditions on the accessory parameters, namely

$$\tilde{c}_i = \bar{c}_i, \qquad i = 1, \ldots, n \tag{3.56}$$

$$\text{Re}\left( \sum_{i=1}^{n} c_i \right) = 0 \tag{3.57}$$

$$\text{Re}\left( \sum_{i=1}^{n} (c_i z_i + \epsilon_i) \right) = 0 \tag{3.58}$$

$$\text{Re}\left( \sum_{i=1}^{n} (c_i z_i^2 + 2\epsilon_i z_i) \right) = 0 \tag{3.59}$$

Upon imposing these we are left with a $2n - 3$ real-dimensional space of undetermined accessory parameters.

**Liouville monodromy problem**

The Liouville solution is determined by a multivalued function $f(z)$ as in (3.34). The Schwarzian derivative of this function is proportional to the stress tensor, cfr. (3.35).

In practice, the function $f(z)$ is constructed from solutions to the ordinary differential equation (ODE)

$$(\partial_z^2 + T(z))\psi(z) = 0. \tag{3.60}$$

Letting $\psi_1$ and $\psi_2$ denote solutions to with unit Wronskian, i.e. $\psi_1'\psi_2 - \psi_2'\psi_1 = 1$, the function $f$ is the ratio

$$f = \frac{\psi_1}{\psi_2}. \tag{3.61}$$

We can also write (3.34) in terms of $\psi_{1,2}$ as

$$e^\Phi = i(\psi_1\bar{\psi}_2 - \bar{\psi}_1\psi_2) \tag{3.62}$$

For general accessory parameters, the vector $(\psi_1 \ \psi_2)^T$ comes back to itself modulo an $SL(2,\mathbb{C})$ monodromy matrix when encircling a singular point, while $f$ transforms by the associated fractional linear transformation. However, the expression (3.34) is only invariant under the $SL(2,\mathbb{R})$ subgroup of $SL(2,\mathbb{C})$. Therefore, in order for $e^\Phi$ to be single-valued, the accessory parameters must be chosen so as to restrict all monodromies to lie in $SL(2,\mathbb{R}) \subset SL(2,\mathbb{C})$. It can be shown [7] that this requirement imposes $2n-3$ real conditions, precisely as many as the number of undetermined accessory parameters. Even though there is no known proof of existence and uniqueness for the Liouville solution with pseudosphere boundary conditions (as far as we know), one therefore expects that at least for some values of the parameters a solution can exist. Below we will derive a more detailed existence condition in the form of a reflection property of a conformal block on the sphere.

**Polyakov relation on the pseudosphere**

As it turns out, the solution of the above monodromy problem, if it exists, is fully determined by the on-shell Liouville action. In the case of Liouville theory on the sphere, this is a well-known property which follows from relations originally conjectured by Polyakov and subsequently proven in [9, 10]. We now show that also on the pseudosphere a Polyakov-type relation holds in the sense that the on-shell Liouville action is a generating function for the accessory parameters:

$$\frac{\partial S_L}{\partial z_i} = 2c_i, \qquad \frac{\partial S_L}{\partial \bar{z}_i} = 2\tilde{c}_i \tag{3.63}$$

The factor of 2 in these expressions can be thought of as coming from contributions from the image charges. We include the derivation here for completeness since we are not aware of its appearance elsewhere. We follow closely the method used in [10] for Liouville theory on the sphere. We want to compute

$$\lim_{\epsilon \to 0} \partial_{z_i} S_L^\epsilon, \tag{3.64}$$

where we recall (cfr. (3.53)) that $S_L^\epsilon$ is given by

$$S_L^\epsilon[\Phi] = \frac{i}{2\pi} \left( \int_{\overline{\mathbb{C}}_\epsilon} dz \wedge d\bar{z} \left( |\partial_z \Phi|^2 - 2e^{-2\Phi} \right) + r_\epsilon \right. \tag{3.65}$$

$$- \sum_{j=1}^n \frac{\alpha_j}{2} \left( \oint_{C_j^\epsilon} \Phi \left( \frac{d\bar{z}}{\bar{z} - \bar{z}_j} - \frac{dz}{z - z_j} \right) + \oint_{\tilde{C}_j^\epsilon} \Phi \left( \frac{d\bar{z}}{\bar{z} - z_j} - \frac{dz}{z - \bar{z}_j} \right) \right) \tag{3.66}$$

$$\left. + \left( \int_{\mathbb{R}+i\epsilon} + \int_{\mathbb{R}-i\epsilon} \right) \Phi \frac{d\bar{z} + dz}{z - \bar{z}} \right), \tag{3.67}$$

with $r_\epsilon$ given in (3.47). In computing the derivative with respect to $z_i$ we have to take into account contributions coming from varying the integration domain, which can be converted into derivatives of step functions $\theta(|z - z_i| - \epsilon)$ in the integrand. This leads to the identity

$$\partial_{z_i} \int_{\overline{\mathbb{C}}_\epsilon} dz \wedge d\bar{z}\, G = \oint_{C_i^\epsilon} G\, d\bar{z} - \oint_{\tilde{C}_i^\epsilon} G\, dz + \int_{\overline{\mathbb{C}}_\epsilon} dz \wedge d\bar{z}\, \partial_{z_i} G. \tag{3.68}$$

Also, the Liouville equation implies that, on $\overline{\mathbb{C}}_\epsilon$,

$$\partial_{z_i} \left( |\partial_z \Phi|^2 + e^{-2\Phi} \right) = \partial_z \left( \partial_{z_i} \Phi \partial_{\bar{z}} \Phi \right) + \partial_{\bar{z}} \left( \partial_{z_i} \Phi \partial_z \Phi \right). \tag{3.69}$$

Making use of these identities we find

$$\partial_{z_i} S_L^\epsilon = \frac{i}{2\pi} \left( \oint_{C_i^\epsilon} \left( |\partial_z \Phi|^2 + e^{-2\Phi} \right) d\bar{z} - \oint_{\tilde{C}_i^\epsilon} \left( |\partial_z \Phi|^2 + e^{-2\Phi} \right) dz \right.$$

$$+ \sum_{j=1}^n \left( \oint_{C_j^\epsilon} + \oint_{\tilde{C}_j^\epsilon} \right) \partial_{z_i} \Phi \left( \partial_{\bar{z}} \Phi\, d\bar{z} - \partial_z \Phi\, dz \right)$$

$$- \sum_{j=1}^n \frac{\alpha_j}{2} \oint_{C_j^\epsilon} \left( \partial_{z_i} \Phi + \delta_{ij} \partial_z \Phi \right) \left( \frac{d\bar{z}}{\bar{z} - \bar{z}_j} - \frac{dz}{z - z_j} \right)$$

$$- \sum_{j=1}^n \frac{\alpha_j}{2} \oint_{\tilde{C}_j^\epsilon} \left( \partial_{z_i} \Phi + \delta_{ij} \partial_{\bar{z}} \Phi \right) \left( \frac{d\bar{z}}{\bar{z} - z_j} - \frac{dz}{z - \bar{z}_j} \right)$$

$$\left. + \left( \int_{\mathbb{R}+i\epsilon} + \int_{\mathbb{R}-i\epsilon} \right) \partial_{z_i} \Phi \frac{d\bar{z} + dz}{z - \bar{z}} \right), \tag{3.70}$$

To evaluate this in the $\epsilon \to 0$ limit we need the first subleading terms in the expansions near the boundary, already derived in (3.16), and near the punctures (3.20). The latter are determined by the form of the stress tensor (3.55) and one finds

$$
\begin{aligned}
\Phi &\overset{z \to z_j}{\sim} \alpha_j \ln|z - z_j| + \sigma_j - \frac{c_j}{\alpha_j}(z - z_j) - \frac{\bar{c}_j}{\alpha_j}(\bar{z} - \bar{z}_j) + \dots, \qquad j = 1, \dots, n \\
&\overset{z \to \bar{z}_j}{\sim} \alpha_j \ln|z - \bar{z}_j| + \tilde{\sigma}_j - \frac{\bar{c}_j}{\alpha_j}(z - \bar{z}_j) - \frac{c_j}{\alpha_j}(\bar{z} - z_j) + \dots, \qquad j = 1, \dots, n \\
&\overset{\operatorname{Im} z \to 0}{\sim} \ln(2\operatorname{Im} z) + \frac{2}{3}(\operatorname{Im} z)^2 T_{|z=\bar{z}} + \dots,
\end{aligned}
\tag{3.71}
$$

where the $\sigma_i$ are functions of $(z_j, \bar{z}_j)$. Using these expansions in (3.70) and taking the $\epsilon \to 0$ limit we obtain

$$
\begin{aligned}
\lim_{\epsilon \to 0} \partial_{z_i} S_L^\epsilon = {}& -c_i \\
& + \sum_{j=1}^{n} \alpha_j (\partial_{z_i}\sigma_j + \partial_{z_i}\tilde{\sigma}_j) + 3c_i \\
& - \sum_{j=1}^{n} \alpha_j \partial_{z_i}\sigma_j \\
& - \sum_{j=1}^{n} \alpha_j \partial_{z_i}\tilde{\sigma}_j \\
& + 0.
\end{aligned} \tag{3.72}
$$

Each line in this expression is the contribution of the corresponding line in (3.70). Adding these up we arrive at (3.63).

## 3.4 Wavefunctions and solutions from vacuum blocks on the sphere

We are now ready to illustrate the intimate connection with classical Virasoro blocks on the sphere. It was argued in [7] that *if* a solution to (2.60) with boundary conditions (3.14) exists, it is closely related to, and determined by, a specific classical Virasoro vacuum block. We will now review and extend this argument. h

### Monodromy problem for classical blocks

We start by recalling some properties of classical Virasoro blocks and their construction through monodromy methods [42], following closely [43]. Quantum conformal blocks $\mathcal{F}(z_I, \Delta_I, \Delta_{J'})$ are basic building blocks of CFT correlators which capture the parts which are fixed by Virasoro symmetry. They depend on the dimensions $\Delta_I, I = 1, \ldots, m$ of the primary operators as well as on dimensions $\Delta_{J'}, J' = 1, \ldots, m-3$ of the exchanged conformal families. They also depend implicitly on the chosen OPE channel in which the exchanged families propagate.

If we take the classical $c = 6k \to \infty$ limit with $\epsilon_I = \Delta_I/k, \nu_{J'} = \Delta_{J'}/k$ fixed, it has been argued that the conformal block exponentiates as follows

$$
\mathcal{F}(z_I, \Delta_I, \Delta_{J'}) \overset{k \to \infty}{\sim} e^{-kF(z_I, \epsilon_I, \nu_{J'})}, \tag{3.73}
$$

where $F$ is called a classical Virasoro block. Let us denote by $D_I$ the partial derivatives

$$
\partial_{z_I} F(z_I, \epsilon_I, \nu_{J'}) \equiv D_I(z_i, \epsilon_i, \nu_{J'}). \tag{3.74}
$$

The conformal Ward identities imply that the $D_I$ play the role of accessory parameters in a meromorphic stress-energy tensor

$$
t(z) = \sum_{I=1}^{m} \left( \frac{\epsilon_I}{(z - z_I)^2} + \frac{D_I}{z - z_I} \right), \tag{3.75}
$$

and satisfy constraints from regularity of $t(z)$ at infinity:

$$\sum_I D_I = 0 \tag{3.76}$$

$$\sum_I (D_I z_I + \epsilon_I) = 0 \tag{3.77}$$

$$\sum_I (D_I z_I^2 + 2\epsilon_I z_I) = 0. \tag{3.78}$$

The classical block accessory parameters $D_I$ are determined [43] by requiring that the solutions to the ordinary differential equation

$$(\partial_z^2 + t(z))\psi = 0 \tag{3.79}$$

have a monodromy matrix $M_{J'}$ when going around the $J'$-th closed loop in the conformal block diagram, whose conjugacy class is fixed by $\nu_{J'}$:

$$\mathrm{tr} M_{J'} = -2\cos(\pi\sqrt{1 - 4\nu_{J'}}). \tag{3.80}$$

**The wavefunction as a classical block**

The following property will allow us to identify the accessory parameters $c_i$ appearing in the Liouville monodromy problem with accessory parameters $D_I$ for a certain classical block. Suppose that a solution to the inhomogeneous Liouville equation (2.60) on the upper half plane exists under boundary conditions (3.14). Then it is easy to see from the reflection property (3.36) that the monodromies of the multivalued function $f$ around image points are each others' inverse:

$$M_{(z_i, \bar{z}_i)} = \left(M_{(\bar{z}_i, z_i)}\right)^{-1}. \tag{3.81}$$

Indeed, let $M$ denote the monodromy matrix that $f$ picks up when encircling the point $(z_i, \bar{z}_i)$ counterclockwise. The reflection property (3.36) then implies that $\bar{M}$ is the monodromy picked up when clockwise encircling the image point $(\bar{z}_i, z_i)$. Since $f$ determines a Liouville solution, all monodromies must lie in $SL(2, \mathbb{R})$ so that $\bar{M} = M$. The monodromy when encircling the image point counterclockwise is therefore $M^{-1}$ as advertised.

The property (3.81) implies that monodromy of solutions to the ODE (3.60), when encircling any pair of image points, is trivial. From the above discussion of classical blocks we then see that the Liouville accessory parameters $c_i, \tilde{c}_i$ solve the monodromy problem determining a $2n$-point conformal block on the sphere. The primary operators are inserted pairwise in image points, and the relevant OPE channel is the one which fuses image pairs, as illustrated in Figure 3. The triviality of the monodromy around image points tells us that this block is a vacuum block. We will use the shorthand notation $F_0(z_i, \bar{z}_i, \epsilon_i)$ to denote this block, i.e.

$$F_0(z_i, \bar{z}_i, \epsilon_i) := F(z_i, z_{n+i} = \bar{z}_i; \epsilon_i, \epsilon_{n+i} = \epsilon_i; \nu_{J'} = 0). \tag{3.82}$$

It generates the Liouville accessory parameters through (3.74),

$$\partial_{z_i} F_0(z_i, \bar{z}_i, \epsilon_i) = c_i, \qquad \partial_{\bar{z}_i} F_0(z_i, \bar{z}_i, \epsilon_i) = \tilde{c}_i. \tag{3.83}$$

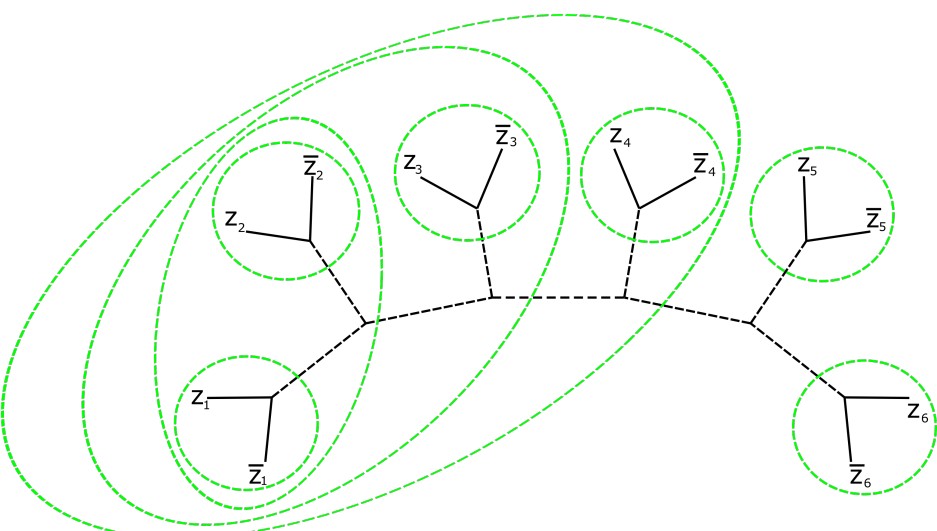

**Figure 3**. The OPE channel relevant for the definition of the vacuum block $F_0$. Solid lines indicate external mirrored operators which are fused together to an identity operator. The green dashed circles indicate cycles of trivial monodromy.

Since the doubling trick for the pseudosphere requires that $\tilde{c}_i = \bar{c}_i$ (see (3.56)), we see that a necessary condition for the Liouville solution to exist is that the block $F_0(z_i, \bar{z}_i, \epsilon_i)$ is, up to an irrelevant constant independent of the insertion points, a real function,

$$F_0(z_i, \bar{z}_i, \epsilon_i) \in \mathbb{R}. \tag{3.84}$$

Note that the remaining conditions on the accessory parameters $c_i$ (3.57-3.59) are then implied by (3.76-3.78).

Let us analyze the existence criterion (3.84) in more detail. In general the conformal block, whether semi-classical or quantum, is a function purely of the cross-ratios, scaling weights, exchange conformal weights and central charge. As is well-known the conformal block in a fixed channel can be constructed level-by-level in terms of descendent states.

In the case of the four-point function of primary operators $\langle \phi_1(0)\phi_2(x)\phi_3(1)\phi_4(\infty) \rangle$ the conformal block for an exchanged primary of holomorphic weight $\Delta_p$ takes the form

$$\mathcal{F}_{34}^{12}(x; p) = \sum_{K=0}^{\infty} \sum_{\{k\}} \beta_{12p}^{\{k\}} x^K \frac{\langle [L, ...[L, \phi_p(0)]...]\phi_3(1)\phi_4(\infty) \rangle}{\langle \phi_p(0)\phi_3(1)\phi_4(\infty) \rangle}. \tag{3.85}$$

Here the sum over $\{k\}$ indicates a sum over all partitions of the integer $K$, the numbers $\beta_{12p}^{\{k\}}$ are the components of the linear decomposition of the OPE into descendent states. Both the numbers $\beta_{12p}^{\{k\}}$ as well as the ratio of 3pt.-functions are fixed entirely by the Virasoro algebra and furthermore they can be constructed entirely by algebraic operations on the scaling dimensions and central charge, as a result these factors are manifestly real. As a consequence, the burden for the reality condition lies on the conformal cross-ratio $x$: as long as $x$ is positive and real the conformal block will be real. For operators inserted in

image points as in (3.82), the conformal cross-ratio takes the form

$$x = \frac{(z_1 - \bar{z}_1)(\bar{z}_2 - z_2)}{(z_1 - \bar{z}_2)(\bar{z}_1 - z_2)}, \tag{3.86}$$

which is manifestly real and positive. Hence in the case of four-points the reality condition imposes no additional constraints.

The situation changes once one considers six points. In this case the relevant cross-ratios are given by

$$z = \frac{(z_5 - z_6)(z_1 - z_4)}{(z_4 - z_5)(z_6 - z_1)}, \quad u = \frac{(z_5 - z_6)(z_2 - z_4)}{(z_4 - z_5)(z_6 - z_2)}, \quad v = \frac{(z_5 - z_6)(z_3 - z_4)}{(z_4 - z_5)(z_6 - z_3)}. \tag{3.87}$$

In the OPE-channel where operators are contracted with the ones located at their mirror points, the so-called 'star'-channel in the case of six points [44], the conformal block takes the form

$$\mathcal{F}^{12}_{3456}(z, u, v; p, q, s) = \sum_{K_1, K_2, K_3} \sum_{\{k_1\}, \{k_2\}, \{k_3\}} z^{K_1} u^{K_2} v^{K_3} \beta^{\{k_1\}}_{12p} \beta^{\{k_2\}}_{34q} \beta^{\{k_3\}}_{56s}$$
$$\times \frac{\langle [L, ...[L, \phi_p(0)]...][L, ...[L, \phi_q(1)]...][L, ...[L, \phi_s(\infty)]...] \rangle}{\langle \phi_p(0)\phi_q(1)\phi_s(\infty) \rangle}. \tag{3.88}$$

By the same argument as before the burden for reality lies on the conformal cross-ratios, the exception now being that the reflection symmetry alone (i.e. $z_4 = \bar{z}_1$, $z_5 = \bar{z}_2$, $z_6 = \bar{z}_3$) is not sufficient to ensure that all three cross-ratios are real and positive. In conclusion, for 3 or more particles on the pseudoshere the existence criterion (3.84) can be met only for the restricted particle positions for which all cross-ratios are real and positive.

Returning to the property (3.83), comparing it with the Polyakov relation (3.63) we see that the Liouville action and the classical block must be proportional (up to an irrelevant additive constant):

$$S_L[\Phi_{cl}(z_i, \bar{z}_i, \alpha_i)] = 2F_0(z_i, \bar{z}_i, \epsilon_i). \tag{3.89}$$

Combining with a similar result for the $\tilde{\Phi}$ Liouville field and our expression (3.54) for the wavefunction in terms of the Liouville action, we obtain the following expression for the classical approximation to the wavefunction (3.39):

$$\Psi_{HH}(z_i, \bar{z}_i, \alpha_i) \sim N(\alpha_i, \tilde{\alpha}_i) e^{-\frac{k}{2}(F_0(z_i, \bar{z}_i, \alpha_i) + F_0(z_i, \bar{z}_i, \tilde{\alpha}_i))}. \tag{3.90}$$

Furthermore, the knowledge of the classical block $F_0$ also determines, in principle, the full (2+1)D backreacted metric: the metric on $\Sigma_0$ is constructed from solutions to two ODE's of the form (3.60) with coefficients determined by $F_0$, and its subsequent time evolution is determined by the first order system (2.36).

## 3.5 Further developments

Before discussing special cases and explicit examples, we would like to clarify two aspects of our formalism. The first is how our framework is related to the description of (2+1)D gravity in terms of a single Liouville field living on the boundary. The second is a concrete description of the improper gauge transformations which 'add boundary gravitons' to the solution.

**Reconstruction of the Lorentzian boundary Liouville field**

It is well-known from the work [45] of Coussaert, Henneaux and van Driel that (2+1)D AdS gravity in the presence of a single conformal boundary can be reformulated in terms of a Lorentzian[12] Liouville field $\Phi_B$ living on the boundary. In this paragraph we wish to clarify the relation between this description and our parametrization in terms of two Euclidean Liouville fields $\Phi, \tilde{\Phi}$ defined on the initial slice $\Sigma_0$.

The boundary Liouville field $\Phi_B$ of [45] satisfies the Lorentzian equation

$$\partial_+ \partial_- \Phi_B + e^{-2\Phi_B} = 0, \tag{3.91}$$

where $x_+, x_-$ are boundary lightcone coordinates. A solution to this equation determines an asymptotically AdS gravity solution whose boundary stress tensor components (rescaled by a factor $-(2k)^{-1}$)) are

$$T(x_+) = -\left((\partial_+ \Phi_B)^2 + \partial_+^2 \Phi_B\right), \qquad \tilde{T}(x_-) = -\left((\partial_- \Phi_B)^2 + \partial_-^2 \Phi_B\right) \tag{3.92}$$

The general solution to (3.91) can be expressed in terms of two real functions $f_B(x_+)$ and $\tilde{f}_B(x_-)$ as

$$e^{-2\Phi_B}(x_+, x_-) = -\frac{f_B'(x_+)\tilde{f}_B'(x_-)}{(f_B(x_+) - f_B(x_-))^2}. \tag{3.93}$$

The associated boundary stress tensors are

$$2T(x_+) = S(f_B, x_+), \qquad 2\tilde{T}(x_-) = S(\tilde{f}_B, x_-). \tag{3.94}$$

In our parametrization, working in the in the upper half plane model for $\Sigma_0$, the bulk Liouville solutions $\Phi$ and $\tilde{\Phi}$ are determined by holomorphic functions $f(z)$ and $\tilde{f}(\bar{z})$ on the upper half plane, which take real values on the real line. Comparing the boundary stress tensors (2.35) and (3.92) in both descriptions shows that we can identify

$$f_B(x_+) = f(x_+), \qquad \tilde{f}_B(x_-) = \tilde{f}(x_-). \tag{3.95}$$

We should note that this simple relation encodes a highly nonlocal map between the Liouville fields $\Phi, \tilde{\Phi}$ and $\Phi_B$. From the boundary point of point of view, the problem of finding the backreacted solution in the presence of particle sources reduces to finding real functions $f_B$ and $\tilde{f}_B$ which extend to multivalued functions $f(z), \tilde{f}(\bar{z})$ on the upper half plane which solve the monodromy problem for Liouville theory on a pseudosphere.

**Boundary gravitons and circle diffeomorphisms**

So far we have discussed how to construct solutions corresponding to particles backreacting on the AdS vacuum solution from conformal blocks for correlators of primaries. In (2+1)D gravity, it is always to generate new solutions by 'adding boundary gravitons', i.e. by performing an improper diffeomorphism which preserves the Brown-Henneaux boundary conditions yet acts nontrivially on the boundary. These diffeomorphisms are the classical

---

[12]See [46, 47] for the role of boundary Liouville theory in Euclidean spacetinmes.

equivalent of acting on the state with a Virasoro group element, and are parametrized by (two copies of) the group of diffeomorphisms of the circle. As discussed in Section, these need to be included in a complete description of the infinite-dimensional phase space, in casu $\mathcal{T}(0, 1, n) \times \mathcal{T}(0, 1, \tilde{n})$.

In our formalism, adding boundary gravitons works as follows. We parametrize the circle by an angular coordinate $\phi$, and consider a general diffeomorphism of $S^1$. The latter can be Fourier expanded as follows

$$\phi \rightarrow \phi' = \phi + \sum_{n \in \mathbb{Z}} a_n e^{in\phi}, \qquad a_{-n} = \overline{a_n} \tag{3.96}$$

To this diffeomorphism we associate a function $g(w)$ as

$$g(w) = z e^{i \sum_{n \in \mathbb{Z}} a_n w^n}. \tag{3.97}$$

The function $g(w)$ is holomorphic on the unit disk and reduces to the diffeomorphism (3.96) on the boundary. Furthermore, it satisfies the reflection condition

$$g(1/w) = \frac{1}{\bar{g}(w)}. \tag{3.98}$$

From our expression (B.11) we see that $g(w)$ is a conformal transformation on $\Sigma_0$ (in the unit disk model) which, acting on a Liouville solution $\Phi$, generates a new solution preserving the ZZ boundary condition. Similarly, we can apply a second conformal transformation of this type to the Liouville solution $\tilde{\Phi}$, and combining these we obtain a (2+1)D gravity solution with added boundary gravitons.

In the simplest cases, where $n = 0, 1$, the incorporation of boundary gravitons reproduces the expected phase space. Indeed, due to the symmetries of the unpunctured and once-punctured hyperbolic disk, the above diffeomorphisms lead to a family of solutions labeled by $\text{Diff}(S^1)/SL(2, \mathbb{R})$ for $n = 0$ and $\text{Diff}(S^1)/U(1)$ for $n = 1$, in agreement with the known phase spaces $\mathcal{T}(0, 1, 0)$ and $\mathcal{T}(0, 1, 1)$. It would be interesting to establish if, for two or more punctures, we similarly obtain a local parametrization of $\mathcal{T}(0, 1, n)$.

## 3.6 Special cases and examples

We conclude our study of spacetimes with a single asymptotic boundary with a discussion of some special classes of solutions and some concrete examples.

### Solutions with only chiral and anti-chiral particles

As anticipated in section (2.4), the class of solutions for which our construction becomes most tractable is that where all the particles are either chiral or antichiral, i.e. $\alpha_i \tilde{\alpha}_i = 0, \forall i$. In this case we can satisfy (2.56) by taking $V$ and $\tilde{V}$ to be as in the vacuum AdS solution and choosing the worldlines of the (and-)chiral particles to be integral curves of the vector field $\partial_t + V$ (resp. $\partial t + \tilde{V}$). For example, in the upper half plane model for $\Sigma_0$, we take $V = -\tilde{V} = -\partial_z - \partial_{\bar{z}}$ and the (anti-) chiral particles move on leftmoving (rightmoving) curves of constant $x_+ = \operatorname{Re} z + t$ (resp. const $x_- = \operatorname{Re} z - t$) at constant values of $y = \operatorname{Im} z$.

In this case the time dependence of the solution is quasi-trivial, with the metric taking the form (3.9). The fields $\Phi$ and $\tilde{\Phi}$ should of course satisfy the sourced Liouville equations (2.33) and obey ZZ boundary conditions (3.14).

Let us work out the explicit solution in the simplest case of one chiral particle and one anti-chiral one, which has not yet appeared in the literature. We take the chiral particle to be of strength $\alpha = 1 - a$ and to start from position $z_0$ at $t = 0$, and the anti-chiral particle of strength $\tilde{\alpha} = 1 - \tilde{a}$ to start from position $\tilde{z}_0$. The conditions (3.56-3.59) fix the accessory parameters in the Liouville stress tensors, which read

$$T(z) = \frac{1 - a^2}{4} \frac{(z_0 - \bar{z}_0)^2}{(z - z_0)^2 (z - \bar{z}_0)^2}, \qquad \tilde{T}(z) = \frac{1 - \tilde{a}^2}{4} \frac{(\tilde{z}_0 - \bar{\tilde{z}}_0)^2}{(z - \tilde{z}_0)^2 (z - \bar{\tilde{z}}_0)^2} \tag{3.99}$$

The corresponding holomorphic functions $f, \tilde{f}$ satisfying (3.35) are

$$f = \left( \frac{z - \bar{z}_0}{z - z_0} \right)^a, \qquad \tilde{f} = \left( \frac{z - \bar{\tilde{z}}_0}{z - \tilde{z}_0} \right)^{\tilde{a}}, \tag{3.100}$$

and the Liouville fields are given through (3.34) by

$$e^{-2\Phi} = a^2 |z_0 - \bar{z}_0|^2 \frac{(|z - z_0||z - \bar{z}_0|)^{2(1-a)}}{4 \left( \mathrm{Im} \left( \frac{z - \bar{z}_0}{z - z_0} \right)^a \right)^2}, \qquad e^{-2\tilde{\Phi}} = \tilde{a}^2 |\tilde{z}_0 - \bar{\tilde{z}}_0|^2 \frac{(|z - \tilde{z}_0||z - \bar{\tilde{z}}_0|)^{2(1-\tilde{a})}}{4 \left( \mathrm{Im} \left( \frac{z - \bar{\tilde{z}}_0}{z - \tilde{z}_0} \right)^a \right)^2}. \tag{3.101}$$

We should note that we have implicitly assumed that the particles move in different planes of constant $\mathrm{Im}z$, i.e. $\mathrm{Im}z_0 \neq \mathrm{Im}\tilde{z}_0$. Otherwise they pass through each other at some time in the future or past, which produces a singularity that we will not analyze here.

**Perturbative solution for a second Liouville source**

The above explicit example required solving Liouville's equation with a single delta-function source under ZZ boundary conditions. For the case of two or more chiral (or non-chiral) particles, we would need to know the Liouville solution with two or more sources, or equivalently a classical sphere block with at least four insertions. Since this function is not known in closed form we will, following [41], construct a perturbative solution in the regime that one of the sources is much weaker than the other. We summarize the computation of [41] (see also [7]) here, which will allow us to perform an important check of our derived relation (3.83) to vacuum blocks on the sphere.

It is convenient to work in the Poincaré disk coordinate $w$, see (3.23). We choose to place the heavier particle with strength $\alpha := 1 - a$ in $w = 0$, and the lighter one of dimension $\epsilon \ll 1$ at $w = r$ with $r$ real. We expand the Liouville stress tensor as

$$T = T_0 + \epsilon T_1 + \mathcal{O}(\epsilon^2) \tag{3.102}$$

$$T_0 = \frac{1 - a^2}{4z^2} \tag{3.103}$$

$$T_1 = \frac{1}{(z - r)^2} + \frac{1}{(z - 1/r)^2} + \frac{d_0}{z} + \frac{d_r}{z - r} + \frac{d_{\frac{1}{r}}}{z - 1/r}. \tag{3.104}$$

The conditions on the accessory parameters (B.13-B.15) can be used to solve for for $d_0$ and $d_{\frac{1}{r}}$, leading to

$$T_1 = \frac{\left(r - \frac{1}{r}\right)^2}{(z-r)^2 \left(z - \frac{1}{r}\right)^2} + \frac{2r - d_r(1-r^2)}{z(z-r)\left(z - \frac{1}{r}\right)},$$

(3.105)

and in addition they imply that $d_r$ is real.

One then proceeds to solve the ordinary differential equation (3.60) to first order in $\epsilon$. We are interested in the monodromy $\delta M$ picked up when encircling the point $z = r$; this can be shown to be [41]

$$\delta M_i^j = 2\pi i \epsilon^{jk} \left(d_r \psi_i^0(r)\psi_k^0(r) + \psi_i^{0\prime}(r)\psi_k^0(r) + \psi_i^0(r)\psi_k^{0\prime}(r)\right),$$

(3.106)

where $\psi_i^0$ are the zeroth-order solutions

$$\psi_1^0 = \frac{1}{\sqrt{a(1-|z_0|^2)}}(z-z_0)^{\frac{1+a}{2}}(1-\bar{z}_0 z)^{\frac{1-a}{2}}$$

(3.107)

$$\psi_2^0 = \frac{1}{\sqrt{a(1-|z_0|^2)}}(z-z_0)^{\frac{1-a}{2}}(1-\bar{z}_0 z)^{\frac{1+a}{2}}.$$

(3.108)

Evaluating (3.106) gives

$$\delta M_1^1 = -\delta M_2^2 = \frac{2\pi i}{a}(1 + rd_r)$$

(3.109)

$$\delta M_1^2 = \frac{-2\pi i r^a}{a}(a + 1 + rd_r)$$

(3.110)

$$\delta M_2^1 = \frac{-2\pi i r^{-a}}{a}(a - 1 - rd_r).$$

(3.111)

To obtain a single-valued Liouville field we should impose that $\delta M$ belongs to $SU(1,1) \subset SL(2,\mathbb{C})$. This leads to

$$\delta M_i^{\ i} = 0$$

(3.112)

$$\delta M_2^2 = \overline{\delta M_1^1}$$

(3.113)

$$\delta M_2^1 = \overline{\delta M_1^2}.$$

(3.114)

The first and second conditions are automatically satisfied (recalling that $d_r$ is real), while the third one determines $d_r$ to be

$$d_r = -\frac{1}{r}\left(1 + a\frac{r^a + r^{-a}}{r^a - r^{-a}}\right).$$

(3.115)

We can now verify that our main identity (3.83) relating the Liouville accessory parameters to classical blocks holds in this example. The classical four-point block was calculated in the same perturbative approximation in [48]:

$$F_0(x) = \left(\ln x + 2\ln \frac{x^{-\frac{a}{2}} - x^{\frac{a}{2}}}{a}\right)\epsilon$$

(3.116)

where $x$ is the crossratio

$$x = \frac{(z_1 - z_2)(z_3 - z_4)}{(z_1 - z_3)(z_2 - z_4)}. \tag{3.117}$$

For the configuration of interest, the insertion points are

$$z_1 = \infty, \qquad z_2 = \frac{1}{r}, \qquad z_3 = r, \qquad z_4 = 0, \tag{3.118}$$

and the crossratio is $x = r^2$. The accessory parameter corresponding to the $z_3$ insertion is

$$c_3 = F'(x)\frac{\partial x}{\partial z_3} \tag{3.119}$$

$$= \left(1 + a\frac{x^{\frac{a}{2}} + x^{-\frac{a}{2}}}{x^{\frac{a}{2}} - x^{-\frac{a}{2}}}\right)\left(\frac{1}{z_1 - z_3} + \frac{1}{z_3 - z_4}\right)\epsilon. \tag{3.120}$$

In particular, for the configuration (3.118) we find

$$c_3 = d_r\epsilon \tag{3.121}$$

with $d_r$ given in (3.115). This confirms our basic property (3.83).

**A scaling limit**

The 2D hyperbolic metrics $ds_2^2$ and $d\tilde{s}_2^2$ that we introduced in Section 2 are generically auxiliary objects which are not embedded in the (2+1)D geometry in any simple way (see (2.20)). There is however a certain scaling limit in which the (2+1)D geometry becomes a fibration over a hyperbolic base manifold with metric $ds_2^2$. As we shall presently see, the limit corresponds to zooming in on a small region of $d\tilde{s}_2^2$ such that the geometry becomes approximately flat. This makes contact with [7, 49] where such limiting solutions were explored in detail.

In order to zoom in on an approximately flat (with possible conical singularities from the sources) region of $d\tilde{s}_2^2$, we perform a scaling limit in which the potential term in the Liouville equation can be neglected. For this purpose we make a field redefinition

$$\tilde{\Phi} \to \tilde{\Phi}' + \Lambda, \tag{3.122}$$

and then taking the $\Lambda \to \infty$ limit while keeping $\Phi'$ and all other variables fixed. The new field satisfies Poisson's equation

$$\partial_z\partial_{\bar{z}}\tilde{\Phi}' = \pi\sum_{i=1}^{n}\tilde{\alpha}_i\delta^{(2)}(z - z_{\tilde{i}}). \tag{3.123}$$

and the metric $d\tilde{s}_2^{2\prime} = e^{-2\Phi'}$ is locally flat. For solutions containing only chiral and antichiral particles, performing the scaling limit in the (2+1)D metric (3.9) leads to

$$ds^2 = e^{-2\Phi(z_+,\bar{z}_+)}dz_+d\bar{z}_+ - \left[\text{Im}\left(\partial_{z_+}\Phi(z_+,\bar{z}_+)dz_+ - \partial_{z_-}\tilde{\Phi}'(z_-,\bar{z}_-)dz_-\right)\right]^2, \tag{3.124}$$

One can verify that this satisfies Einstein's equations. Note that attempting to take a similar scaling limit on both $\Phi$ and $\tilde{\Phi}$ would lead to a degenerate metric.

In the absence of antichiral sources, $\tilde{\alpha}_i = 0 \ \forall i$, we can take the following solution for $\tilde{\Phi}$:

$$\tilde{\Phi}'(z, \bar{z}) = \mathrm{Im}\ z. \tag{3.125}$$

The metric then becomes

$$ds^2 = e^{-2\Phi(z_+, \bar{z}_+)}dz_+ d\bar{z}_+ - \left( dt - \mathrm{Im}\left( \left( \frac{i}{2} + \partial_{z_+}\Phi(z_+, \bar{z}_+) \right) dz_+ \right) \right)^2. \tag{3.126}$$

In this limit, the (2+1)D geometry takes the form of a timelike fibration over a hyperbolic base manifold with metric $ds_2^2$. This particular scaling limit for solutions with chiral particles was studied extensively in [7, 49]. We note that, as $\tilde{A}_\tau \sim L_0$, our criterion (2.62) is obeyed and the chiral particles move on geodesics. This was also checked explicitly in [7].

## 4 Towards holography for closed universes

So far we have considered a time slice $\Sigma_0$ with an asymptotic boundary, as appropriate for describing particles in an asymptotically AdS spacetime. However our formalism applies in principle to time slices of arbitrary topology, and it is interesting to consider the case where $\Sigma_0$ is a compact Riemann surface without boundary. The corresponding (2+1)D solutions are closed universes evolving from a big bang to big a crunch singularity and were considered in a related context in [19, 20], see also [50] for a review and further references. The study of such boundary-less slices generalizes the standard AdS/CFT setup and could be a step towards studying cosmological singularities.

If $\Sigma_0$ is a surface of genus $g$, the Gauss-Bonnet theorem places a necessary condition on the parameters $0 \le \alpha_i < 1$ for a solution to exist [51]:

$$\sum_{i=1}^n \alpha_i > 2(1 - g). \tag{4.1}$$

For simplicity we assume in what follows that $\Sigma_0$ has spherical topology, $g = 0$. Note that we need $n \ge 3$ to satisfy (4.1). The classical result going back to Picard [52] is that, when (4.1) is obeyed, a solution to the inhomogeneous Liouville equation (2.60) exists and is unique.

As we did in the case with asymptotic boundary, we will study a semiclassical Hartle-Hawking path integral preparing the multi-particle state on $\Sigma_0$, which will again be closely related to the Liouville action on $\Sigma_0$. We then go on to examine the role of classical Virasoro blocks in determining this wavefunction and the gravity solution.

### 4.1 Hartle-Hawking wavefunction and Liouville action

We are interested in computing a Chern-Simons path integral, which is analogous to the Hartle-Hawking [35] path integral over metrics, and prepares a multi-particle state on the Riemann sphere $\Sigma_0 = \overline{\mathbb{C}}$ at $t = 0$. We take the particles to be located at $z_i, i = 1, \dots, n$ with quantum numbers $\alpha_i, \tilde{\alpha}_i$. By applying Möbius isometries of $\Sigma_0$ we can fix the last three particle locations to

$$z_{n-2} = 0, \qquad z_{n-1} = 1, \qquad z_n = \infty. \tag{4.2}$$

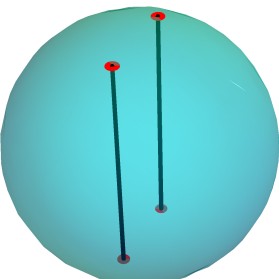

**Figure 4**. The lower half of Figure 1(c) can be redrawn as a ball with particle worldlines connecting points on the boundary two-sphere through the bulk.

Near the sources, the Liouville field $\Phi$ has the asymptotics

$$\Phi \overset{z \to z_j}{\sim} \alpha_j \ln|z - z_j| + \mathcal{O}(1), \qquad j = 1, \ldots, n - 1 \tag{4.3}$$

$$\overset{z \to \infty}{\sim} (2 - \alpha_n) \ln|z| + \mathcal{O}(1), \tag{4.4}$$

and similarly for $\tilde{\Phi}$.

In analogy with the Hartle-Hawking no-boundary proposal we want to perform a path integral on a 3-manifold $X$ whose boundary at $t = 0$ is $\Sigma_0$ and without boundaries in the past. Each particle location $z_i$ is the endpoint of a worldline in $X$. Since our framework is not equipped to deal with worldlines ending on each other[13] - for one thing, our gauge choice would break down at such an endpoint - each worldline needs to connect two particle locations on $\Sigma_0$. We therefore restrict attention to configurations of particles at $\Sigma_0$ which come in pairs with the same quantum numbers $\alpha_i, \tilde{\alpha}_i$. The manifold $X$ is then a ball containing particle worldlines which connect pairs of boundary points (see Figure 4).

Proceeding as in Section 3.2, we find that the total action evaluates to a boundary term which reads

$$iS_{CS}^{tot} = -\frac{k}{4} S_L[\Phi] + \frac{3ik}{8\pi} \lim_{\epsilon \to 0} \int_{\overline{\mathbb{C}}_\epsilon} dz \wedge d\bar{z} e^{-2\Phi}, \tag{4.5}$$

---

[13]To describe more general configurations, we need a framework where worldlines can end on each other, in other words which incorporated bulk interactions. This is beyond the scope of the present work, see however [53].

where $S_L[\Phi]$ is the standard, regularized, Liouville action [54]:

$$S_L[\Phi] = \lim_{\epsilon \to 0} S_L^\epsilon[\Phi]$$

$$S_L^\epsilon[\Phi] = \frac{i}{2\pi} \left( \int_{\overline{\mathbb{C}}_\epsilon} dz \wedge d\bar{z} \left( |\partial_z \Phi|^2 + e^{-2\Phi} \right) + r_\epsilon \right.$$

$$- \sum_{j=1}^{n-1} \frac{\alpha_j}{2} \oint_{C_j^\epsilon} \Phi \left( \frac{d\bar{z}}{\bar{z} - \bar{z}_j} - \frac{dz}{z - z_j} \right)$$

$$\left. - \left( 1 - \frac{\alpha_n}{2} \right) \oint_{C_n^\epsilon} \Phi \left( \frac{d\bar{z}}{\bar{z}} - \frac{dz}{z} \right) \right). \tag{4.6}$$

Here, $\overline{\mathbb{C}}_\epsilon$ is the extended complex plane with disks of radius $\epsilon$ removed around the insertion points $z_i$ (including $z_n = \infty$) and $C_i^\epsilon$ are the corresponding boundary curves. The constant

$$r_\epsilon = -\pi \left( \sum_{j=1}^{n-1} \alpha_j^2 + (2 - \alpha_n)^2 \right) \ln \epsilon. \tag{4.7}$$

is included to obtain a finite result in the $\epsilon \to 0$ limit.

Using the fact the source-free Liouville equation holds on $\overline{\mathbb{C}}_\epsilon$ and the asymptotic behavior (4.3,4.4), the last term can be evaluated to yield

$$iS_{CS}^{tot} = -\frac{k}{4} S_L[\Phi] - \frac{3k}{8} \left( \sum_{j=1}^{n} \alpha_j - 2 \right). \tag{4.8}$$

Therefore, the on-shell Chern-Simons action is once again proportional to the on-shell Liouville action, up to an additive constant depending only on the quantum numbers $\alpha_i$ of the particles but not on their locations $z_i$.

## 4.2 Liouville action, accessory parameters and classical blocks

Let us review the Polyakov relations satisfied by the on-shell Liouville action, as well as its interpretation in terms of the geometry of Teichmüller space. As in the case with conformal boundary, the problem of solving the sourced Liouville equation can be recast as a monodromy problem for a set of accessory parameters. The Liouville stress tensor (2.35) is a meromorphic function of the form

$$T(z) = \sum_{i=1}^{n-1} \left( \frac{\epsilon_i}{(z - z_i)^2} + \frac{c_i}{z - z_i} \right), \tag{4.9}$$

In addition, from (4.4) the behavior of $T$ near $z \to \infty$ is given by

$$T(z) \sim \frac{\epsilon_n}{z^2} + \frac{c_n}{z^3} + \mathcal{O}(z^{-4}), \qquad z \to \infty. \tag{4.10}$$

The accessory parameters $c_i$ are subject to three three linear relations imposed by the asymptotics (4.10),

$$\sum_{i=1}^{n-1} c_i = 0, \qquad \sum_{i=1}^{n-1} (\epsilon_i + c_i z_i) = \epsilon_n, \qquad \sum_{i=1}^{n-1} (2\epsilon_i z_i + c_i z_i^2) = c_n. \tag{4.11}$$

Therefore the independent accessory parameters can be taken to be $c_1, \ldots, c_{n-3}$.

The Liouville solution is again obtained from solutions to the ODE (3.60), whose monodromies around the singular points should once again lie in the subgroup $SL(2\,\mathbb{R})$ of $SL(2,\mathbb{C})$. One can show (see e.g. [7]) that this imposes $2(n-3)$ real conditions, precisely as many as there are undetermined accessory parameters. The existence and uniqueness of the Liouville solution implies that this monodromy problem indeed uniquely fixes the accessory parameters.

The accessory parameters have an important interpretation in terms of the Kähler geometry of Teichmüller space. It can be shown [9] that they are real-analytic functions of the moduli. They are however not holomorphic, and their antiholomorphic derivatives give the Weil-Petersson metric on Teichmüller space:

$$\frac{\partial c_i}{\partial \bar{z}_j} = g_{i\bar{j}}^{WP}. \tag{4.12}$$

A conjecture by Polyakov, which was rigorously proven in [9, 10], states that the on-shell Liouville action (4.6) is a generating function of the accessory parameters:

$$\frac{\partial S_L}{\partial z_i} = c_i. \tag{4.13}$$

Combining this with (4.12) shows that $S_L$ is a Kähler potential for the Weil-Petersson metric. Due to the property (4.8), this is also the case for the total action $iS_{CS}^{tot}$.

A derivation of Polyakov's conjecture (4.13) along the lines of [10] proceeds as in Section 3.3. We want to compute

$$\lim_{\epsilon \to 0} \partial_{z_i} S_L^\epsilon, \tag{4.14}$$

with $S_L^\epsilon$ given in (4.6). The following identity takes into account the variations of the integration domain $\overline{\mathbb{C}}_\epsilon$

$$\partial_{z_i} \int_{\overline{\mathbb{C}}_\epsilon} dz \wedge d\bar{z}\, G = \oint_{C_i^\epsilon} G\, d\bar{z} + \int_{\overline{\mathbb{C}}_\epsilon} dz \wedge d\bar{z}\, \partial_{z_i} G. \tag{4.15}$$

Making use of (3.68) we find

$$\begin{aligned}
\partial_{z_i} S_L^\epsilon = \frac{i}{2\pi} &\left( \oint_{C_i^\epsilon} \left( |\partial_z \Phi|^2 + e^{-2\Phi} \right) d\bar{z} + \sum_{j=1}^n \oint_{C_j^\epsilon} \partial_{z_i} \Phi \left( \partial_{\bar{z}} \Phi d\bar{z} - \partial_z \Phi dz \right) \right. \\
&- \frac{1}{2} \sum_{j=1}^{n-1} \alpha_j \oint_{C_j^\epsilon} \left( \partial_{z_i} \Phi + \delta_{ij} \partial_z \Phi \right) \left( \frac{d\bar{z}}{\bar{z} - \bar{z}_j} - \frac{dz}{z - z_j} \right) \\
&\left. - \left( 1 - \frac{\alpha_n}{2} \right) \oint_{C_n^\epsilon} \partial_{z_i} \Phi \left( \frac{d\bar{z}}{\bar{z}} - \frac{dz}{z} \right) \right),
\end{aligned} \tag{4.16}$$

where each line in this expression is the derivative of the corresponding line in (4.6). To evaluate this in the $\epsilon \to 0$ limit we need the first subleading terms in the expansions (4.3,

4.4). The form of the stress tensor (4.9) determines these to be of the form

$$\Phi \stackrel{z \to z_j}{\sim} \alpha_j \ln|z - z_j| + \sigma_j - \frac{c_j}{\alpha_j}(z - z_j) - \frac{\bar{c}_j}{\alpha_j}(\bar{z} - \bar{z}_j) + \dots, \qquad j = 1, \dots, n-1$$

$$\stackrel{z \to \infty}{\sim} (2 - \alpha_n)\ln|z| + \sigma_n - \frac{c_n}{\alpha_n}\frac{1}{z} - \frac{\bar{c}_n}{\alpha_n}\frac{1}{\bar{z}} + \dots, \tag{4.17}$$

where the $\sigma_i$ are functions of $(z_j, \bar{z}_j)$. Using these expansions in (4.16) and taking the $\epsilon \to 0$ limit we obtain

$$\begin{aligned}
\partial_{z_i} S_L &= \sum_{j=1}^{n-1} \alpha_j \partial_{z_i}\sigma_j + (\alpha_n - 2)\partial_{z_i}\sigma_n + c_i \\
&\quad - \sum_{j=1}^{n-1} \alpha_j \partial_{z_i}\sigma_j \\
&\quad - (\alpha_n - 2)\partial_{z_i}\sigma_n.
\end{aligned} \tag{4.18}$$

Adding these up we arrive at (4.13).

Having reviewed the Polyakov relation (4.13) we can now relate the accessory parameters $c_i$ to the classical Virasoro blocks. The unique solution to the inhomogeneous Liouville equation determines a stress tensor $T(z)$ whose associated ODE, in some chosen OPE channel, has monodromies corresponding to a specific set of exchanged families $\nu_{i'}^*$. Therefore the $c_i$ should coincide with the accessory parameters in the monodromy problem determining a classical block in the chosen channel. In the notation of section (3.3) we have

$$c_i(z_j, \bar{z}_j) = D_i(z_j, \epsilon_i, \nu_{i'}^*). \tag{4.19}$$

We note that, for consistency with (4.12), the dimensions $\nu_{i'}^*$ must implicitly depend on the $z_j$ and their complex conjugates $\bar{z}_j$. We can say a bit more on this dependence from the known properties of Liouville theory. It can be argued from the large-$k$ behavior of of Liouville correlators that the on-shell action is of the form [54]

$$S_L = \left(\mathcal{S}^{(3)}(\epsilon_i, \nu_{i'}) + F(z_i, \epsilon_i, \nu_{i'}) + \bar{F}(\bar{z}_i, \epsilon_i, \nu_{i'})\right)\Big|_{\nu_{i'} = \nu_{i'}^*} \tag{4.20}$$

Here, $\mathcal{S}^{(3)}$ contains the contribution from three-point coefficients at large-$k$, and the right-hand side is evaluated on exchanged dimensions $\nu_{i'}^*(z_j, \bar{z}_j)$ which extremize this contribution. This property, together with the Polyakov relations (4.13), then implies the relation (4.19), since

$$\frac{\partial S_L}{\partial z_i} = \frac{\partial F(z_i, \epsilon_i, \nu_{i'}^*)}{\partial z_i}. \tag{4.21}$$

To conclude, we have established that the Hartle-Hawking wavefunction for a closed universe containing some point particles is once again determined by the on-shell Liouville action. In contrast to the situation with asymptotic boundary however, the action and accessory parameters contain dynamical information of Liouville theory beyond the kinematical data contained in conformal blocks. This information enters through the nontrivial dependence of the exchanged families on the insertion points $\nu_{i'}^*(z_j, \bar{z}_j)$.

To emphasize the increased challenge associated to the closed surface situation, it is helpful to compare how the Liouville problem on a closed surface and on the pseudosphere problem relate to the conformal block problem of section 3.4. The Liouville monodromy problem requires us to restrict the monodromies to fall within $SL(2,\mathbb{R})$, while in the conformal block problem one fixes a smaller number of monodromies but in addition one fixes the exact conjugacy classes of $SL(2,\mathbb{R})$. In the case of the pseudosphere these problems coincide due to the additional structure provided by the reflection symmetry. This fixes the resulting conformal block to the identity block of the channel depicted in figure 3.

In the closed surface case, where there is no such additional structure, the solution to the Liouville problem will once again provide a conformal block, but which conformal block one obtains is a non-kinematical question that is sensitive to the OPE structure of Liouville theory.

## 5 Outlook

We conclude by listing some open problems and future directions.

- In this work we restricted attention to spacetimes containing point particles but no black holes. It is of obvious interest to generalize our discussion to bulk states containing black holes. A first interesting configuration would be to consider states containing point particle excitations on top of an eternal BTZ black hole (see also [55]). In this case the slice $\Sigma$ would contain a second asymptotic boundary. A second question would be to address dynamical black hole formation from the collapse of point particles [56]. We would for example hope that our framework could make more explicit make the link between conformal blocks with a continuous operator distribution and Vaidya-type metrics proposed in [57].

- Since the considerations in this work were semiclassical, it would be of great interest to extend them to the quantum regime. One of the difficulties in doing so is the quotient by the mapping class group in the gravitational phase space (see Footnote 6), which is hard to implement in the Chern-Simons path integral as pointed out in [18]. It would also be useful to clarify how our semiclassical wavefunctions emerge at large $k$ from those considered in [11], which were derived in the 'quantize first and then constrain' approach.

### Acknowledgements

JR would like to thank Ondra Hulík and Orestis Vasilakis for initial collaboration and valuable discussions. We furthermore thank T. Anous, S. Banerjee, M. Kudrna and T. Procházka for useful discussions. This work was supported by the Grant Agency of the Czech Republic under the grant EXPRO 20-25775X.

## A  MPD equations

The Mathisson-Papapetrou-Dixon (MPD) equations describe the motion of an intrinsically spinning particle in general relativity. They take the form [33]

$$\nabla_s(mu^\mu + u_\nu \nabla_s S^{\mu\nu}) = \frac{1}{2} R^\mu{}_{\nu\rho\sigma} u^\nu S^{\rho\sigma} \tag{A.1}$$

$$\nabla_s S^{\mu\nu} + u^\mu u_\rho \nabla_s S^{\nu\rho} - u^\nu u_\rho \nabla_s S^{\mu\rho} = 0 \tag{A.2}$$

where $S^{\mu\nu}$ is an antisymmetric tensor called the spin tensor and $u^\mu = \frac{dx^\mu}{ds}$ is the velocity of the worldline parametrized by proper time, $u^\mu u_\mu = -1$. The worldline covariant derivative is defined as

$$\nabla_s v^\mu = \frac{dv^\mu}{ds} + \Gamma^\mu_{\nu\rho} u^\nu v^\rho. \tag{A.3}$$

We are interested in the MPD equations in (2+1)D spacetimes which are locally AdS. The right-hand side of (A.1) simplifies in this case to

$$\frac{1}{2} R^\mu{}_{\nu\rho\sigma} u^\nu S^{\rho\sigma} = S^{\mu\nu} u_\nu. \tag{A.4}$$

The MPD equations then allow for special class of solutions where the worldline is a geodesic, i.e.

$$\nabla_s u^\mu = 0, \tag{A.5}$$

One can show that (A.1,A.2) then imply that

$$S^{\mu\nu} = \sigma \epsilon^{\mu\nu\rho} u_\rho, \tag{A.6}$$

where $\sigma$ is an arbitrary proportionality constant.

Following the steps in Appendix E of [31], the MPD equations (A.1,A.2) can be rewritten as

$$\frac{dP}{ds} + [\Omega_s + E_s, P] = 0 \tag{A.7}$$

$$\frac{d\tilde{P}}{ds} + [\Omega_s - E_s, \tilde{P}] = 0 \tag{A.8}$$

where $\Omega_s = \frac{1}{2} \epsilon^a{}_{bc} \Omega_{\mu bc} u^\mu J_a$, $E_s = E^a_\mu u^\mu J_a$, $E$ and $\Omega$ are the (2+1)D vielbein and spin connection, and $J_a$ are $sl(2,\mathbb{R})$ generators satisfying $[J_a, J_b] = \epsilon_{abc} \eta^{cd} J_d$. The quantitites $P$ and $\tilde{P}$ in (A.7,A.8) are defined as

$$P := \left( mu^a + u_b \nabla_s S^{ab} + \frac{1}{2} \epsilon^a{}_{bc} S^{bc} \right) J_a \tag{A.9}$$

$$\tilde{P} := \left( mu^a + u_b \nabla_s S^{ab} - \frac{1}{2} \epsilon^a{}_{bc} S^{bc} \right) J_a. \tag{A.10}$$

The form (A.7,A.8) of the MPD equations precisely coincides with the equations of motion (2.47) for the Chern-Simons-matter system in the main text. Note that (A.7,A.8) imply

that $\mathrm{tr}P^2$ and $\mathrm{tr}\tilde{P}^2$ are constant, so that the additional equation of motion (2.48) serves to fix integration constants in terms of the mass and spin of the particle.

For example, for solutions describing geodesic motion (A.5,A.6), the momenta take the form

$$P = (m + \sigma)u^a J_a, \qquad P = (m - \sigma)u^a J_a \tag{A.11}$$

The equation (2.48) then tells us that $\sigma$ is to be identified with the particle helicity $s$. From (A.5, A.11) we infer that general solutions describing geodesic motion are characterized by

$$[P, \tilde{P}] = \nabla_s P = \nabla_s \tilde{P} = 0, \tag{A.12}$$

or, equivalently, making use of (A.7,A.8),

$$[P, \tilde{P}] = [E_s, P] = [E_s, \tilde{P}] = 0. \tag{A.13}$$

## B  Some formulas for the Poincaré disk model

In this Appendix we collect some formulas relevant when using the Poincaré disk model for $\Sigma_0$. As in (3.23) we use a complex coordinate $w$, $|w| < 1$.

Let us start by deriving the form of the (2+1)D metric. A natural choice for the vector fields $V, \tilde{V}$ specifying the Chern-Simons gauge choice (2.11) is

$$V = -iw\partial_w + i\bar{w}\partial_{\bar{w}}, \qquad \tilde{V} = iw\partial_w - i\bar{w}\partial_{\bar{w}}. \tag{B.1}$$

From our time evolution equations (2.36) we see that the fields $\lambda, \tilde{\lambda}$ should be turned on at times $t \neq 0$, leading to

$$\Phi(t, w, \bar{w}) = \Phi(we^{it}, \bar{w}e^{-it}), \qquad\qquad \tilde{\Phi}(t, w, \bar{w}) = \Phi(we^{-it}, \bar{w}e^{it}) \tag{B.2}$$
$$\lambda = -t, \qquad\qquad \tilde{\lambda} = t \tag{B.3}$$
$$\mu = 0, \qquad\qquad \tilde{\mu} = 0 \tag{B.4}$$

Defining the combinations

$$w_+ = we^{it}, \qquad w_= we^{-it}, \tag{B.5}$$

corresponding 3D metric is

$$ds^2 = \left| e^{-\Phi(w_+, \bar{w}_+)}dz_+ + e^{-\tilde{\Phi}(w_-, \bar{z}_-)}dw_- \right|^2 - \left[ \mathrm{Im}\left( \partial_{w_+}\Phi(w_+, \bar{w}_+)dw_+ - \partial_{w_-}\tilde{\Phi}(w_-, \bar{w}_-)dw_- \right) \right]^2. \tag{B.6}$$

We now summarize some formulas needed for solving the inhomogeneous Liouville equation on the unit disk with ZZ boundary conditions, referring to [7] for more details. Asymptotically AdS solutions are described by Liouville fields with ZZ boundary conditions which take the form

$$e^{2\Phi} = (1 - |w|^2)^2 + \mathcal{O}((1 - |w|^2)^4), \tag{B.7}$$

and the stress tensor obeys

$$\left. \left( w^2 T(w) \right) \right|_{|w|=1} \in \mathbb{R}. \tag{B.8}$$

The doubling trick extending $T$ to the Riemann sphere is

$$T(w) = \frac{1}{w^4}\bar{T}(1/w). \tag{B.9}$$

A general Liouville solution is specified by a function $g(w) = \psi_1(w)/\psi_2(w)$ through

$$e^{-2\Phi} = (\psi_1\bar{\psi}_1 - \psi_2\bar{\psi}_2)^{-2} = \frac{|g'|^2}{(1-|g|^2)^2}, \tag{B.10}$$

and the appropriate doubling trick for $g$ reads

$$g(w) = \frac{1}{\bar{g}(1/w)}. \tag{B.11}$$

Now let us consider the presence of $n$ particle sources in locations $w_i$, where we assume[14] for definiteness that $w_n = 0$. The stress tensor of an inhomogeneous Liouville solution is of the form

$$T(w) = \frac{\epsilon_n}{w^2} + \frac{c_n}{w} + \sum_{i=1}^{n-1}\left(\frac{\epsilon_i}{(w-w_i)^2} + \frac{\tilde{\epsilon}_i}{(w-1/\bar{w}_i)^2} + \frac{c_i}{w-w_i} + \frac{\tilde{c}_i}{w-1/\bar{w}_i}\right), \tag{B.12}$$

where the accessory parameters $c_i, \tilde{c}_i$ are constrained by regularity at infinity and by the reflection condition (B.9) to obey

$$2\epsilon_i + c_i w_i + \frac{\bar{\tilde{c}}_i}{w_i} = 0 \tag{B.13}$$

$$c_n + \sum_{i=1}^{n-1}(c_i + \tilde{c}_i) = 0 \tag{B.14}$$

$$\text{Im}\left(\sum_{i=1}^{n-1}\left(c_i w_i - \frac{\bar{\tilde{c}}_i}{w_i}\right)\right) = -2\epsilon_n. \tag{B.15}$$

The accessory parameters are determined by requiring the monodromy of the ODE (3.60) around each of the sources to lie in $SU(1,1) \subset SL(2,\mathbb{C})$, so that the Liouville field (B.10) is single-valued.

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
