# Peer review of "Holography for bulk states in 3D quantum gravity"

_SciPost Physics_

## Round 1 · Referee Report · Anonymous · 2023-1-2

Report
In this paper the authors discuss the semiclassical wavefunction of 3d gravity and its relation to Liouville CFT and Virasoro conformal blocks. There are a number of nice new results, in particular on properly including spin in the description of conical defects. However there are also several subtle points that have not been addressed sufficiently, including some which may affect the main conclusions. I would recommend this article for acceptance if significant revisions are made to address these points. Let me emphasize that I think this is a nice paper with an interesting perspective on this problem that differs from some of the literature, and some of the comments below may stem from my own confusions.
My comments are lengthy, so here is the short version. In sections 3 and 4 I do not understand (i) exactly what saddlepoint is claimed to have this on-shell action, or (ii) what boundary conditions the authors have used to implicitly define the state "HH" whose wavefunction is calculated. Both of these confusions would be greatly clarified if the authors provided a detailed example where they write down the spacetime metric and boundary conditions at all boundaries (not just $\Sigma_0$). If they do this, I suspect they will find that they have implicitly chosen very special, moduli-dependent boundary conditions in the definition of the HH state.
Requested changes
Major comments/questions are as follows:
1. In section 3, the authors calculate a "wavefunction" $\Psi_{HH}$ by evaluating the on-shell action of a saddlepoint. However some parts of this argument are too quick, so I cannot tell exactly what they claim to be calculating. Some parts of the argument appear to be in Lorentzian and some in Euclidean, with no discussion of the distinction or any Wick rotation (which is very subtle for particles with spin, and has never been addressed fully in the literature).
First, I do not understand what saddlepoint is being analyzed. For concreteness, take $\Sigma_0$ to be a disk with two spinless defects. Is the spacetime Euclidean or Lorentzian? In section 3.2 it is described as a "filled-in pseudosphere." If we construct such a 3-geometry in Euclidean signature, then the pseudosphere itself is the conformal boundary. But I don't think this is what the authors have done, because that would require additional boundary terms at the conformal boundary, and these would contribute to the on-shell action. So I think the authors have in mind a Lorentzian 3-manifold. But if we take the initial data on $\Sigma_0$ and evolve in Lorentzian time, I think the resulting spacetime will be incomplete: these coordinates will cover only the causal diamond associated to $\Sigma_0$. To see this, consider for example the case where $\Sigma_0$ is simply an empty disk with no defects. Then the resulting 3-manifold will be the hyperbolic patch of AdS3, which is a diamond embedded in the global Lorentzian cylinder, hitting the boundary at $t=0$. More generally, why is the wavefunction the action of this diamond? And what state is this the wavefunction of? HH wavefunctions are normally computed by Euclidean path integrals, and the authors use that language, but this does not seems to correspond to the calculation, as it is done in Lorentzian.
To try to make my confusion more clear I will restate it another way: What is the definition of the state $|HH\rangle$ whose wavefunction is being calculated here? A wavefunction is an overlap, schematically $\Psi_{HH}(X) = \langle X| HH\rangle$, where $X$ is the data on $\Sigma_0$. But what boundary conditions are the authors using to define the quantum state $|HH\rangle$? Usually (for example, in the classic paper of Hartle and Hawking) these boundary conditions would be imposed in Euclidean signature. However, if so, then there is additional data that must be provided --- the positions where the defects hit the boundary. Furthermore there will be an additional boundary term at Euclidean time $t_{E} \to -\infty$ that has not been included in the calculation.
In Lorentzian signature we have a similar problem: the Lorentzian path integral calculates an overlap between two states, and the initial state has not been defined. This state will presumably have particles piercing the horizon of the causal diamond, and the locations of those particles are therefore part of the definition of the initial state. Consider, for example, the simple case where $\Sigma_0$ is a disk with $n$ defects. The Lorentzian metric
\begin{align}
ds^2 = -dt^2 + \cos^2 t e^{\Phi}|dz|^2
\end{align}
covers a causal diamond; the particles enter through the lower part of the diamond, and exit through the top. So I do not see how the action of this diamond can be interpreted as a HH wavefunction.
Another related question: Does the state $|HH\rangle$ depend on the moduli, i.e. the particle insertion points $(z_i, \bar{z}_i)$? The notation suggests no, but I think the answer is yes, because the particles hit the boundary of the spacetime. Therefore the state $|HH\rangle$ has implicitly been defined by turning on special $z_i$-dependent sources at either the Euclidean boundary or the Lorentzian $t \to -\infty$ slice.
2. I have some similar questions about section 4. I do not understand the geometry of this saddlepoint, or even its asymptotics. First, I am confused by figure 4. This drawing appears to correspond to taking $\Sigma_0$ to be a sphere with two particles. However, there is no hyperbolic metric on the sphere with two defects, so it is impossible to satisfy the constraint equation. So what then is figure 4 illustrating? (My interpretation of this figure was that the $t=0$ slice $\Sigma_0$ is obtained as the horizontal slice in the middle of the figure, but this leads to the contradiction just stated. Maybe I'm just misreading the figure somehow.)
Perhaps this was an error and the authors meant to have $n \geq 3$ defects, so let's consider the case where $\Sigma_0$ is a sphere with three particles. But then there is another issue. If we take this initial data and evolve it in Lorentzian signature we will find a bang/crunch singularity, as in [Maldacena Maoz hep-th/0401024]. Yet the authors seem to claim there is a smooth 3-manifold. Alternatively, we can evolve in Euclidean signature --- then the 3-manifold is smooth, but it has (at least) two asymptotic conformal boundaries, whereas the authors consider only manifolds without conformal boundary.
The natural analogue of the HH wavefunction for a closed hyperbolic 3-manifold is in fact discussed in the above-mentioned paper by Maldacena and Maoz. But it comes from a Euclidean path integral with two disjoint conformal boundaries. If we allow the particles to hit the boundary, then there is additional data in the definition fo the state. But if we do not allow the particles to hit the boundary, then I do not think it will be possible to satisfy the constraint equations.
I am therefore skeptical that there is any smooth (away from defects) boundaryless manifold of the type discussed in section 4 that satisfies the constraints.
And some additional minor comments:
3. There is a large literature on the relation between the large-c Virasoro identity block and 3d gravity with defects that is neither discussed nor cited. Some discussion of the existing literature should be added, including citations of the original papers showing this relation which are [Faulkner 1303.7221] and [Hartman 1303.6955] (the latter paper is cited, but not for this). In particular the main result (1.3) of the current paper, when specialized to particles without spin, is the "square root" of the equation $Z_{3d gravity} = |{\cal F}|^2$ derived in those papers. Thus for the case of spinless particles, the result (1.3) was already well known (modulo the subtleties raised above about Lorentzian vs Euclidean -- the above literature is in Euclidean signature where there is a clear interpretation as a partition function and wavefunction). The main advance in the current paper is incorporating spinning external operators, and this should be emphasized more clearly.
4. I believe there is a typo in eqn (1.4) in the intro: it should have $k/4 \to k/2$ in the exponent. The factor of $k/4$ disagrees with the factor $k/2$ in eqn (3.90). The $k/2$ factor is the correct one, because if we set the spins to zero and square this equation, we must recover the result $Z_{grav} = |{\cal F}|^2$ obtained in [1303.7221,1303.6955].
5. The introductions says the paper will only consider "states containing a number of point-like particles, but no black holes. " On the contrary, the geometries discussed in this paper do include black holes in many cases! For example if we insert two scalar particles with $h > c/32$, then the resulting spatial slice $\Sigma_0$ has a nontrivial geodesic surrounding them. Therefore this is a black hole in a pure state, with two particles behind the horizon. I assume what the authors' meant was that they do not consider thermal states, or eternal black holes, as that would require multiple boundaries.
6. There are several typos: On p.8 "correct correct". On p. 20 "is cannot completely arbitrary" and capitalization of ". the". On p.25, a lonely "h". On p.28 "pseudoshere". On p.29 "in the in the".
Author: Gideon Vos on 2023-02-10 [id 3341]
(in reply to Report 1 on 2023-01-02)
We would like to thank the referee for their valuable comments:
1) A first point we should clarify is that in this work we don't perform a standard Hartle-Hawking path integral over metrics, where as the Referee points out, it would indeed be crucial to continue to Euclidean signature. Instead, we perform here a somewhat analogous Hartle-Hawking-like path integral preparing a state in the SL(2,R) Cherns-Simons theory. In topological theories, where the Hamiltonian vanishes (modulo subtleties if a boundary is present), it is not necessary to continue to imaginary time as both $e^{ i t H}$ and $e^{ - \tau H}$ are the identity. State-preparing path intergals in Chern-Simons theory, of the kind we consider, have also appeared in the literature without the need for continuation to imaginary time, see for example ref. [36]. We have indicated what we mean by a `Hartle-Hawking-like' path integral in the Introduction and included an extensive discussion at the start of section 3.2.
A second important point is that, as the Referee rightly points out, our definition of the HH-like state does include a very specific set of boundary conditions on the past boundary. (Such a dependence on past boundary conditions is standard in HH path integrals in spacetimes with conformal boundary as explained e.g. in ref. [37].) In our case of interest, we include sources on the past boundary which are located precisely in the image points of the insertions on the unit disk at $t=0$. We have stressed the dependence on past boundary sources in the second paragraph of Section 3.2 and also below (110). We also agree with the Referee that the action should include appropriate boundary terms on the past boundary. These are indeed included in our discussion: they are the contributions to our boundary action (120-122) coming from the lower half plane and the image charges located there.
2) We believe that this comment was likely due to a misinterpretation of Figure 4: the constant time slice is here a two-sphere with 4 particle insertions. As in the HH no-boundary proposal, on performs a path integral by filling in the two-sphere, obtaining a ball pierced by particle worldlines. We have clarified the caption to Figure 4 so as to avoid confusion.
3) and 4) We would like to thank the Referee for this insighful comment, which gives a consistency check on the numerical factors in our calculation which we had not realized. The mod-square of the HH wavefunction is expected to represent a correlation function on the boundary sphere and our result (161) (or (3), there was indeed (there was indeed a factor 2 missing in this latter equation) then reproduces the result in 1303.7221,1303.6955 for nonspinning sources, and generalizes it to the spinning case. We have commented on this check in the Introduction and at the end of Section 3.4, as well as included references to 1303.7221,1303.6955.
5) The referee is correct, this particular abuse of terminology has now been addressed on page 2.
6) We have corrected these typos.
Author: Gideon Vos on 2023-02-10 [id 3340]
(in reply to Report 2 on 2023-01-19)From a pure AdS/CFT dictionary perspective it might be surprising that the gravitational state is given by a CFT conformal block, since as the referee rightly points out is a single term attributed to a CFT correlator rather than a CFT state. In the language of gravitational state preparation this is quite conventional, in the works [13,18] that follow the quantize-first-constrain-later approach to quantization, the exact operator constraint imposed takes the form of the Virasoro ward identity, hence the constrained wavefunctionals take the form of conformal blocks.
In the boundary CFT, the state in the Hilbert space that we are considering should be obtainable by performing a path integral on the hemispherical 'cap' in Fig. 1(b), with primary operators inserted at various points. Through the OPE, this can also be viewed as the state created by a single, highly non-primary, operator at the south pole of the cap. We have included a footnote to this effect in the Introduction p. 4.
Also, the typo's that were pointed out have been corrected

---

## Round 1 · Referee Report · Anonymous · 2023-1-19

Strengths
1- Paper is self contained and exhaustive
2- Builds on a program of both the authors
3- Nicely follows through on providing a picture relating bulk states to the CFT Hilbert space in the spirit of CFT/AdS
4-The authors have done a good job distilling difficult-to-read results from past literature and making it accessible
Weaknesses
1- Just a few typos
Report
This is a nice paper that follows through on the promise of finding a direct link between the quantum gravity Hilbert space in AdS$_3$ with point particle sources, and states in the CFT. This paper is restricted to the semiclassical picture, and relates the quantum state wavefunction on a Cauchy slice to the Virasoro vacuum block on the sphere.
It would be amazing to understand if this program can be utilized to understand the crossover into the quantum regime where the Chern-Simons scaled to be small. I very much hope the authors continue along this line of research.
I am generally confused about the point being made about the bulk Hartle-Hawking wavefunction and the Hilbert space, which I'm sure the authors could clear up. Namely the CFT Hilbert space is spanned by the local operators inserted at the origin. However this bulk wavefunction is an element of the bulk Hilbert space on the slice, but it seems to be spanned by a number of particles in the bulk and its insertion. Its relation to the vacuum block makes it seem much closer to a CFT correlation function rather than a CFT state. This subtlety is perhaps only confusing to me, but I think if the authors could clear it up, that would be very helpful.
Requested changes
I would like some general comments on how I should think about the Hilbert space in the bulk, or if present already in the paper, just refer me to the discussion already present.
In addition, I found a few typos (but this list is by no means exhaustive):
1. In equation (2.25) the right equation should have an $\tilde{\omega}$
2. off $\rightarrow$ of after equation (2.38)
3. A trailing "h"at the end of the paragraph in section 3.4 should be removed

---

## Editorial Decision

submission_&_refereeing_history